# Centriolar satellites expedite mother centriole remodeling to promote ciliogenesis

Emma A Hall[1†], Dhivya Kumar[2†], Suzanna L Prosser[3], Patricia L Yeyati[1], Vicente Herranz-Pérez[4,5], Jose Manuel García-Verdugo[4], Lorraine Rose[1], Lisa McKie[1], Daniel O Dodd[1], Peter A Tennant[1], Roly Megaw[1], Laura C Murphy[6], Marisa F Ferreira[1], Graeme Grimes[1], Lucy Williams[1], Tooba Quidwai[1], Laurence Pelletier[3,7], Jeremy F Reiter[2,8*], Pleasantine Mill[1*]

[1]MRC Human Genetics Unit, Institute of Genetics and Cancer, University of Edinburgh, Edinburgh, United Kingdom; [2]Department of Biochemistry and Biophysics, Cardiovascular Research Institute, University of California, San Francisco, United States; [3]Lunenfeld-Tanenbaum Research Institute, Sinai Health System, Toronto, Canada; [4]Cavanilles Institute of Biodiversity and Evolutionary Biology, University of Valencia, Valencia, Spain; [5]Predepartamental Unit of Medicine, Jaume I University, Castelló de la Plana, Spain; [6]Institute of Genetics and Cancer, University of Edinburgh, Edinburgh, United Kingdom; [7]Department of Molecular Genetics, University of Toronto, University of Toronto, Canada; [8]Chan Zuckerberg Biohub, San Francisco, United States

*For correspondence:
Jeremy.Reiter@ucsf.edu (JFR);
pleasantine.mill@ed.ac.uk (PM)

†These authors contributed equally to this work

**Abstract** Centrosomes are orbited by centriolar satellites, dynamic multiprotein assemblies nucleated by Pericentriolar material 1 (PCM1). To study the requirement for centriolar satellites, we generated mice lacking PCM1, a crucial component of satellites. *Pcm1−/−* mice display partially penetrant perinatal lethality with survivors exhibiting hydrocephalus, oligospermia, and cerebellar hypoplasia, and variably expressive phenotypes such as hydronephrosis. As many of these phenotypes have been observed in human ciliopathies and satellites are implicated in cilia biology, we investigated whether cilia were affected. PCM1 was dispensable for ciliogenesis in many cell types, whereas *Pcm1−/−* multiciliated ependymal cells and human *PCM1−/−* retinal pigmented epithelial 1 (RPE1) cells showed reduced ciliogenesis. *PCM1−/−* RPE1 cells displayed reduced docking of the mother centriole to the ciliary vesicle and removal of CP110 and CEP97 from the distal mother centriole, indicating compromised early ciliogenesis. Similarly, *Pcm1−/−* ependymal cells exhibited reduced removal of CP110 from basal bodies in vivo. We propose that PCM1 and centriolar satellites facilitate efficient trafficking of proteins to and from centrioles, including the departure of CP110 and CEP97 to initiate ciliogenesis, and that the threshold to trigger ciliogenesis differs between cell types.

## Editor's evaluation

This manuscript will be of interest to centrosome and cilia cell biologists and evaluates the in vivo and in vitro role of PCM1, and by extension, centriole satellites in ciliogenesis. The major strength of this study is the detailed characterization of *Pcm1−/−* mutant mice, which reveals a role for PCM1 in the biogenesis of specific types of cilia, such as motile cilia on ependymal cells, by a mechanism involving CP110 and CEP97. The claims are generally well supported by the data.

## Introduction

A pair of microtubule-based centrioles form the heart of the centrosome. In addition to roles in spindle formation during mitosis, centrioles are critical to ciliogenesis, the process of building a cilium during interphase (*Nigg and Holland, 2018*). In most cells, the older mother centriole uniquely matures into the basal body, which serves as the foundation for the primary cilium, a single signaling antenna. In contrast, multiciliated cells lining the trachea, oviduct, and brain ventricles generate many basal bodies that then nucleate many motile cilia per cell.

In all cells, dynamic remodeling of centrioles is required for ciliogenesis. Key early steps in ciliogenesis include basal body acquisition of distal appendages and the removal of CP110 and CEP97 from the distal end of the mother centriole, two proteins that inhibit assembly of the ciliary axoneme (*Čajánek and Nigg, 2014*; *Goetz et al., 2012*; *Schmidt et al., 2012*; *Schmidt et al., 2009*; *Sillibourne et al., 2013*; *Spektor et al., 2007*; *Tanos et al., 2013*; *Tsang et al., 2008*). How the cell controls centriole remodeling remains unclear.

Surrounding the centrosome and ciliary base are centriolar satellites, small membrane-less granules which move along cytoplasmic microtubules (*Bärenz et al., 2011*; *Kubo et al., 1999*; *Kubo and Tsukita, 2003*; *Odabasi et al., 2020*). PCM1 is both a component of centriolar satellites and necessary for centriolar satellite formation (*Dammermann and Merdes, 2002*; *Kubo and Tsukita, 2003*; *Odabasi et al., 2019*; *Wang et al., 2016*). With PCM1, a diverse array of proteins co-localize at centriolar satellites (*Gheiratmand et al., 2019*; *Gupta et al., 2015*; *Odabasi et al., 2019*; *Quarantotti et al., 2019*), and many of these components also localize at centrioles themselves (*Kodani et al., 2015*; *Lopes et al., 2011*). Centriolar satellites are dynamic, change in response to cell stresses, and have been implicated in diverse processes including Hedgehog signaling, autophagy, proteasome activity, and aggresome formation (*Holdgaard et al., 2019*; *Hori and Toda, 2017*; *Joachim et al., 2017*; *Kubo and Tsukita, 2003*; *Lecland and Merdes, 2018*; *Odabasi et al., 2019*; *Prosser et al., 2022*; *Prosser and Pelletier, 2020*; *Tang et al., 2013*; *Tollenaere et al., 2015*; *Villumsen et al., 2013*; *Wang et al., 2013a*). Possibly reflecting involvement in these diverse biological processes, genetic perturbation of centriolar satellite components can compromise cilia formation and contribute to human ciliopathies and microcephaly (*Conkar et al., 2017*; *Kim et al., 2008*; *Klinger et al., 2014*; *Kurtulmus et al., 2016*; *Lee and Stearns, 2013*; *Mikule et al., 2007*; *Staples et al., 2014*), perhaps in a cell-type-specific way (*Monroe et al., 2020*; *Odabasi et al., 2019*; *Wang et al., 2016*). Thus, understanding of the function of PCM1 and centriolar satellites is emerging.

To investigate the functions of centriolar satellites in vivo, we generated *Pcm1* null mice. We found that PCM1 is important for perinatal survival. *Pcm1*$^{-/-}$ mice surviving the perinatal period displayed dwarfism, male infertility, hydrocephaly, cerebellar hypoplasia, and variably expressive ciliopathy-associated phenotypes such as hydronephrosis, reflecting important roles for centriolar satellites in promoting both primary and motile ciliogenesis. In assessing how centriolar satellites enable ciliogenesis, we found that cells lacking PCM1 display compromised docking of the mother centriole to the ciliary vesicle and attenuated removal of CP110 and CEP97. Thus, we propose that centriolar satellites shape the mother centriole to promote critical early steps in ciliogenesis.

## Results

### *Pcm1*$^{-/-}$ mice exhibit perinatal lethality and ciliopathy-associated phenotypes

To investigate the in vivo function of centriolar satellites in mammals, we used CRISPR/Cas9 to create deletions in mouse *Pcm1*. Among the mutations generated, *Pcm1*$^{\Delta 5\text{-}14}$ introduced a frameshift after the first amino acid leading to a premature stop and *Pcm1*$^{\Delta 796\text{-}800}$ caused a frameshift and premature stop in exon 6 (*Figure 1—figure supplement 1A*). Immunoblotting with antibodies to two regions of PCM1, PCM1 immunofluorescence of mouse embryonic fibroblasts (MEFs) derived from *Pcm1* mutant mice, and mass spectrometry-based proteomic analysis indicated that both mutations prevented formation of detectable PCM1 protein (*Figure 1A, B*, *Figure 1—figure supplement 1B, C*). Mice homozygous for either *Pcm1* mutation exhibited indistinguishable phenotypes (*Figure 1—figure supplement 2*). Thus, we surmise that both mutations are likely to be null and henceforth we refer to both alleles as *Pcm1*$^{-}$.

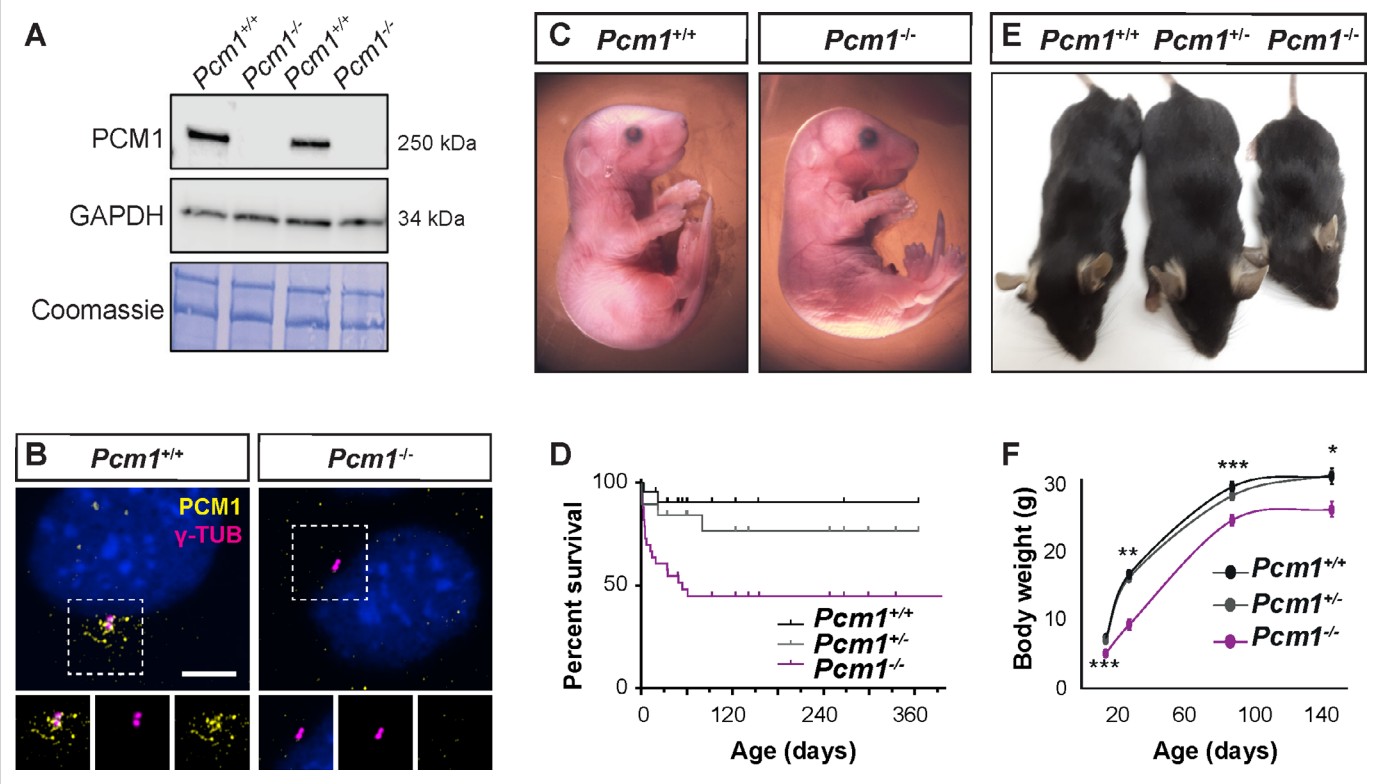

**Figure 1.** PCM1 is important for perinatal survival. (**A**) Immunoblot of mouse embryonic fibroblast (MEF) lysates from wild-type and $Pcm1^{-/-}$ MEFs for PCM1 and GAPDH (loading control). Gel stained with Coomassie blue. (**B**) Immunostaining of PCM1 (yellow) and centrioles (γ-tubulin, γ-TUB, magenta) in wild-type and $Pcm1^{-/-}$ MEFs. (**C**) E18.5 wild-type and $Pcm1^{-/-}$ neonates. (**D**) Kaplan–Meier curve of wild-type, $Pcm1^{+/-}$ and $Pcm1^{-/-}$ mice. See also *Figure 1—figure supplement 1D*. (**E**) P28 wild-type, $Pcm1^{+/-}$ and $Pcm1^{-/-}$ mice. (**F**) Graph of body weights of wild-type, $Pcm1^{+/-}$ and $Pcm1^{-/-}$ mice by age. Student's *t*-test *$p < 0.05$, **$p < 0.01$, ***$p < 0.001$. Error bars represent standard error of the mean (SEM), $n > 7$ per genotype at P14 and $n > 3$ per genotype at P150.

The online version of this article includes the following source data and figure supplement(s) for figure 1:

**Source data 1.** Full uncropped immunoblots for *Figure 1A* and *Figure 1—figure supplement 1C*, labeled and unlabeled.

**Figure supplement 1.** PCM1 promotes survival and growth.

**Figure supplement 2.** $Pcm1^{\Delta5\text{-}14/\Delta5\text{-}14}$ and $Pcm1^{\Delta796\text{-}800/\Delta796\text{-}800}$ mice exhibit comparable phenotypes.

**Figure supplement 3.** PCM1 is dispensable for ciliogenesis in some cell types.

$Pcm1^{-/-}$ mice were present at normal Mendelian ratios at late gestation (embryonic day [E] 18.5) (*Figure 1C, D*, *Figure 1—figure supplement 1D*). As abrogation of cilia themselves results in midgestation lethality (*Huangfu et al., 2003*), the presence of $Pcm1^{-/-}$ embryos late in gestation suggests that PCM1 is not essential for all ciliogenesis. Indeed, cilia in several $Pcm1^{-/-}$ tissues were morphologically normal at E18.5 (*Figure 1—figure supplement 3A–E*). However, by postnatal day (P) 5, half of $Pcm1^{-/-}$ mice had died (*Figure 1D*, *Figure 1—figure supplement 1D*), revealing that PCM1 is important for perinatal survival.

Surviving $Pcm1^{-/-}$ mice were smaller than littermate controls, weighing less than half of controls at P28 (*Figure 1E, F*). This dwarfism was detectable before birth, indicating intrauterine growth retardation (*Figure 1—figure supplement 1E*). The brains of surviving $Pcm1^{-/-}$ mice were proportionally smaller than those of littermates (*Figure 1—figure supplement 1F, G*), and displayed marked hydrocephaly (*Figure 2A–C*, *Figure 2—figure supplement 1A, B*). Hydrocephaly can result from motile cilia dysfunction, raising the possibility that centriolar satellites are required for cilia formation and/or function in ependymal cells.

In the postnatal brain, primary cilia are critical for Hedgehog signaling in cerebellar granule cell precursors. Decreased cerebellar Hedgehog signaling attenuates expansion of the granule cell precursors (*Dahmane and Ruiz i Altaba, 1999*; *Spassky et al., 2008*; *Wallace, 1999*; *Wechsler-Reya*

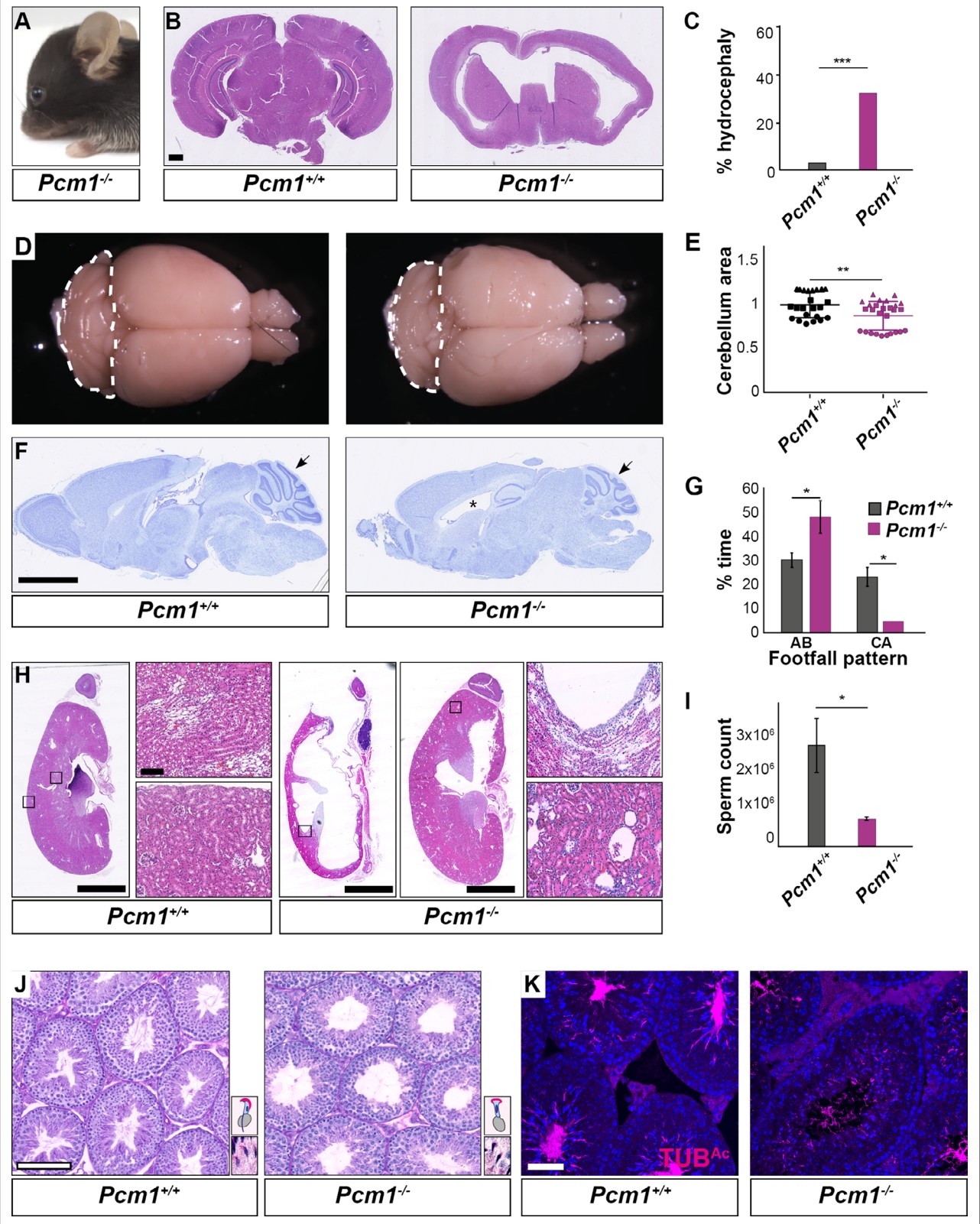

**Figure 2.** *Pcm1*−/− mice display ciliopathy-associated phenotypes. (**A**) *Pcm1*−/− mouse displaying a domed skull indicative of hydrocephaly. (**B**) Coronal sections of 5-week-old wild-type and *Pcm1*−/− brains. (**C**) Percentages of wild-type and *Pcm1*−/− mice exhibiting hydrocephaly (*n* = 22 *Pcm1*−/− mice, age 19 days to 3 months (with hydrocephaly) and 6 weeks to 1 year (without overt hydrocephaly), *n* = 35 age-matched littermate controls). ***p < 0.001 (**D**) Gross morphology of 8-month-old wild-type and *Pcm1*−/− brains. Cerebella are delineated with dotted lines. (**E**) Quantification of cerebellar

*Figure 2 continued on next page*

*Figure 2 continued*

area measured from sagittal sections of 2- to 8-month-old brains from $Pcm1^{-/-}$ mice without frank hydrocephaly, normalized to the mean of wild-type cerebellar area. N = 3. Each shape represents a different animal. Error bars indicate standard deviations. Student's *t*-test: \*\*p < 0.01. (**F**) Cresyl violet-stained sagittal sections of 8-month-old brains. Cerebella are indicated with arrows. \*Dilated ventricle. (**G**) Percentage of time spent by adult wild-type and $Pcm1^{-/-}$ mice in alternate (AB) gait and cruciate (CA) gait. Mean ± standard error of the mean (SEM). $Pcm1^{+/+}$ n = 4, $Pcm1^{-/-}$ n = 5. Student's *t*-test: \*p < 0.05. (**H**) H&E-stained sections of kidneys and adrenals from 6-week-old wild-type and $Pcm1^{-/-}$ mice. (**I**) Sperm count per ml of wild-type and $Pcm1^{-/-}$ epididymal semen. n = 3 per genotype. Error bars represent SEM. Student's *t*-test: \*p < 0.05 (**J**) PAS-stained sections of 3-month-old wild-type and $Pcm1^{-/-}$ seminiferous tubules. Insets are higher magnification images of elongated spermatids (see *Figure 2—figure supplement 1C* for lower magnification images), with a cartoon of sperm head morphology. (**K**) Immunofluorescence staining of wild-type and $Pcm1^{-/-}$ seminiferous tubules for sperm flagella (acetylated tubulin, $TUB^{Ac}$, magenta) and nuclei (DAPI, blue). Scale bars represent 1 mm in **B**, 2.5 mm in **F** and **H**, 100 μm in **J**, and 50 μm in **K**.

The online version of this article includes the following video and figure supplement(s) for figure 2:

**Figure supplement 1.** $Pcm1^{-/-}$ mice display a subset of ciliopathy-associated phenotypes.

**Figure 2—video 1.** Wild-type sperm morphology and movement.

https://elifesciences.org/articles/79299/figures#fig2video1

**Figure 2—video 2.** $Pcm1^{-/-}$ sperm are immotile and lack normal head structures.

https://elifesciences.org/articles/79299/figures#fig2video2

**Figure 2—video 3.** $Pcm1^{-/-}$ sperm exhibit disrupted movement.

https://elifesciences.org/articles/79299/figures#fig2video3

*and Scott, 1999*). The cerebella of $Pcm1^{-/-}$ mice were smaller than those of littermate controls (*Figure 2D–F*, *Figure 2—figure supplement 1A*). As the cerebellum is important for motor coordination, we analyzed the gait of surviving $Pcm1^{-/-}$ mice. Consistent with altered cerebellar function, $Pcm1^{-/-}$ mice displayed ataxia (*Figure 2G*).

We investigated whether $Pcm1^{-/-}$ mice exhibit other Hedgehog-associated phenotypes. A proportion of viable $Pcm1^{-/-}$ mice (n = 2/15) developed hydronephrosis (*Figure 2H*), which can also result from attenuated Hedgehog signaling (*Yu et al., 2002*).

Because retinal degeneration is characteristic of several ciliopathies and PCM1 was strongly expressed in the retina (*Figure 2—figure supplement 1D*), we examined the retinas of $Pcm1^{-/-}$ mice using fundal imaging and histological analysis at 1 year of age. $Pcm1^{-/-}$ mice did not display characteristic features of photoreceptor death, such as changes to retinal pigmentation on fundoscopy or reduction of the outer nuclear layer on histology (*Figure 2—figure supplement 1E, F*). Electroretinogram (ERG) testing at 9 months of age revealed no visual functional deficits in $Pcm1^{-/-}$ mice (*Figure 2—figure supplement 1G–I*). Therefore, PCM1 is not essential for photoreceptor survival, suggesting it is dispensable for photoreceptor ciliogenesis and ciliary trafficking.

Surviving $Pcm1^{-/-}$ male mice were infertile with reduced sperm in seminiferous tubules (*Figure 2I–K*, *Figure 2—figure supplement 1C*). The few $Pcm1^{-/-}$ sperm identified exhibited disrupted head-to-tail coupling, abnormal head morphology indicative of defective intramanchette trafficking, and immotility (*Figure 2—figure supplement 1C*, *Figure 2—videos 1–3*). We previously discovered similar defects in male mice lacking centriolar satellite component CEP131 (also known as AZI1) (*Hall et al., 2013*), consistent with the idea that centriolar satellites are essential for mammalian spermatogenesis and male fertility. Thus, PCM1 supports postnatal survival and is required for the function of multiple ciliated cell types.

## PCM1 promotes ciliogenesis in multiciliated cells

During the perinatal period, ependymal cells lining the brain ventricles generate many motile cilia. Shortly after birth (P1), immature ependymal cells possess non-polarized, short cilia. Beginning at P3, ependymal cells form multiple long, polarized cilia; this ciliogenesis occurs in a wave across the ventricle from caudal to rostral. By P15, ependymal cilia mature to generate metachronal rhythm (*Spassky et al., 2008*). Recent work showed that knockdown of $Pcm1$ in cultured ependymal cells led to disrupted cilia ultrastructure and motility (*Zhao et al., 2021*).

To explore whether defects in ependymal cilia could be the cause of hydrocephaly in $Pcm1^{-/-}$ mice, we imaged ependymal cilia in lateral ventricle walls. $Pcm1^{-/-}$ mice exhibited numerous ependymal cell abnormalities, including fewer ependymal cells with multiple basal bodies at P3 and P5 (*Figure 3A-C*, *Figure 3—figure supplement 1A*). However, by P16, $Pcm1^{-/-}$ mice had caught up and displayed

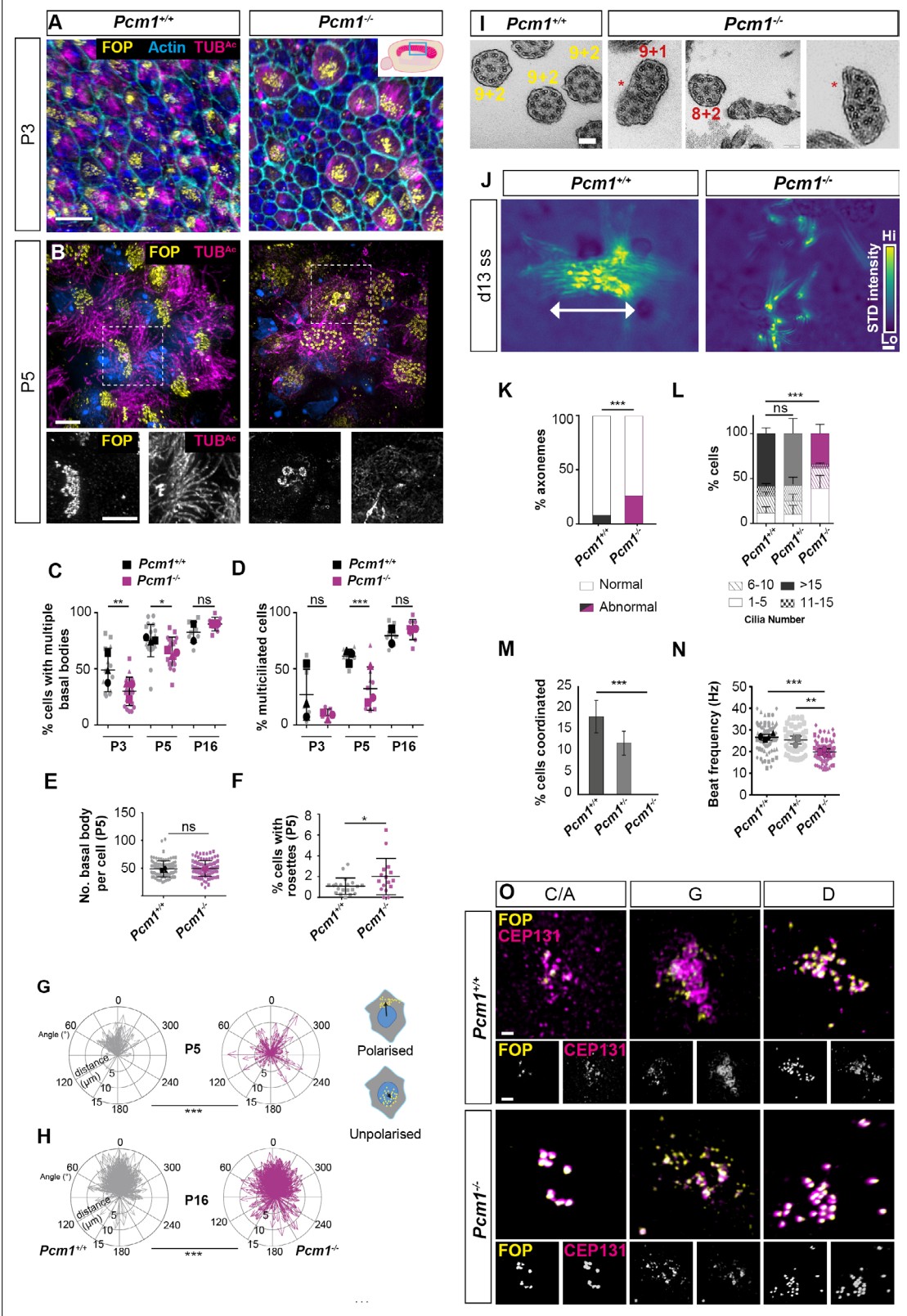

**Figure 3.** PCM1 is required for efficient basal body synthesis and multiciliogenesis. (**A**) Wild-type and *Pcm1⁻/⁻* P3 wholemount brain ventricles immunostained for basal bodies (FOP, yellow), actin (phalloidin, cyan) and cilia (TUB^Ac, magenta). Inset depicts area of ventricle imaged (cyan box). (**B**) Wild-type and *Pcm1⁻/⁻* P5 wholemount ventricles immunostained for basal bodies (FOP, yellow), cilia (TUB^Ac, magenta), and nuclei (DAPI, blue). Below: single optical planes highlight the persistence of rosettes and disrupted ciliogenesis in *Pcm1⁻/⁻* ependymal cells. (**C**) Percentage of ependymal cells with

*Figure 3 continued on next page*

*Figure 3 continued*

>4 basal bodies in wild-type and *Pcm1*⁻/⁻ P3, P5, and P16 ventricles. Each shape represents an animal; the smaller symbols represent individual images and the larger shape the mean for each animal. Student's *t*-test: *p < 0.05, **p < 0.01, ns: not significant. (**D**) Percentage of ependymal cells with multiple cilia in wild-type and *Pcm1*⁻/⁻ P3, P5, and P16 ventricles. Student's *t*-test: ***p < 0.001, ns, not significant. (**E**) The number of basal bodies per wild-type and *Pcm1*⁻/⁻ P5 ependymal cell. (**F**) Percentage of P3 wild-type and *Pcm1*⁻/⁻ ependymal cells with centriolar rosette structures. Student's *t*-test: *p < 0.05. (**G, H**) Rose plots of the translational polarity of basal bodies in wild-type and *Pcm1*⁻/⁻ P5 and P16 ependymal cells, as assessed from immunofluorescent images as in *Figure 3—figure supplement 1A, B*. Schematic insets represent individual ependymal cells with polarized or unpolarized basal bodies (yellow). An arrow was drawn from the center of the nucleus (blue) to the center of the basal bodies (yellow) and the distance and angle is plotted relative to the average angle for that field of view, which was set to 0˚. At both P5 and P16, the standard deviations between wild-type and *Pcm1*⁻/⁻ ependymal cells are different (*F*-test: ***p < 0.0001). (**I**) Transmission electron microscopy (TEM) of ependymal cell cilia from P3 wild-type and *Pcm1*⁻/⁻ ventricles. Wild-type cilia display 9 + 2 microtubule arrangement. *Pcm1*⁻/⁻ cilia display axonemal defects, including missing microtubule doublets and axoneme fusion (indicated by *). (**J**) Colorized heat map (scale: yellow – high, blue – low) of maximum projection of the standard deviation of pixel intensity in *Figure 3—videos 1 and 2*, depicting wild-type and *Pcm1*⁻/⁻ cultured ependymal cell cilia beat coordination. Areas of high pixel intensity variation reflect areas of increased movement. (**K**) Percentage of P3 wild-type and *Pcm1*⁻/⁻ ependymal cilia structural anomalies. Chi-squared test: ***p < 0.001. *n* = 121 cilia from 3 wild-type mice and 61 cilia from 3 *Pcm1*⁻/⁻ mice. (**L**) Percentage of cultured wild-type and *Pcm1*⁻/⁻ ependymal cells with ranges of cilia number 14–16 days after serum withdrawal. Chi-squared test: ***p < 0.001. ns: not significant. (**M**) Percentage of cultured wild-type and *Pcm1*⁻/⁻ ependymal cells with coordinated ciliary beating 14–16 days after serum withdrawal. Chi-squared test: ***p < 0.01. (**N**) Cilia beat frequency of cultured wild-type and *Pcm1*⁻/⁻ ependymal cells 14–16 days after serum withdrawal. Small symbols represent individual cells, large symbols represent average for each cell lines from an individual animal. Student's *t*-test: ***p < 0.001, **p < 0.01. (**O**) Representative images of wild-type and *Pcm1*⁻/⁻ mouse tracheal epithelium cells (mTECs) cultured at air–liquid interface for 3 days and immunostained for basal bodies (FOP, yellow) and CEP131 (magenta). Representative cells cultured from *n* = 3 wild-type and 3 *Pcm1*⁻/⁻ animals, at the 'centriolar amplification' (C/A), 'growth' (G), and 'disengagement' (D) stages of centriolar amplification are shown (see also *Figure 3—figure supplement 1G*). Scale bars: 15 µm (**A**), 5 µm (**B**), 100 nm (**I**), and 1 µm main panel, 2 µm inset (**O**). Error bars represent SEM.

The online version of this article includes the following video and figure supplement(s) for figure 3:

**Figure supplement 1.** Centriole amplification is delayed and fibrogranular material is disrupted in *Pcm1*⁻/⁻ ependymal cells.

**Figure supplement 2.** *Pcm1*⁻/⁻ ependymal cells form elongated centriole-like structures.

**Figure supplement 3.** Delayed expression of ciliary proteins in *Pcm1*⁻/⁻ mouse tracheal epithelium cells (mTECs).

**Figure 3—video 1.** Wild-type cultured ependymal cilia beat in a coordinated way 14 days after serum withdrawal.
https://elifesciences.org/articles/79299/figures#fig3video1

**Figure 3—video 2.** *Pcm1*⁻/⁻ ependymal cilia show uncoordinated, slow ciliary beat 16 days after serum withdrawal.
https://elifesciences.org/articles/79299/figures#fig3video2

**Figure 3—video 3.** *Pcm1*⁻/⁻ ependymal cilia show uncoordinated, slow ciliary beat 14 days after serum withdrawal.
https://elifesciences.org/articles/79299/figures#fig3video3

**Figure 3—video 4.** Cilia of a tracheal wholemount preparation from a 3-month-old wild-type mouse beating.
https://elifesciences.org/articles/79299/figures#fig3video4

**Figure 3—video 5.** Cilia of a tracheal wholemount preparation from a 2-month-old *Pcm1*⁻/⁻ mouse beating.
https://elifesciences.org/articles/79299/figures#fig3video5

**Figure 3—video 6.** Wild-type ALI12 mouse tracheal epithelium cell (mTEC) cilia beating.
https://elifesciences.org/articles/79299/figures#fig3video6

**Figure 3—video 7.** *Pcm1*⁻/⁻ ALI12 mouse tracheal epithelium cell (mTEC) cilia beating.
https://elifesciences.org/articles/79299/figures#fig3video7

normal numbers of ependymal cells with multiple basal bodies (*Figure 3C*, *Figure 3—figure supplement 1B*). These results suggest a delay in centriole biogenesis in the absence of PCM1.

Once committed to making multiple centrioles, the numbers of basal bodies per cell formed by *Pcm1*⁻/⁻ ependymal cells in vivo was similar to controls at P5 (*Figure 3B, E*). However, at this early stage, *Pcm1*⁻/⁻ mice also exhibited increased numbers of cells with rosette-like arrangements of basal bodies (*Figure 3B, F*). As rosettes are typically present earlier in ependymal centriole biogenesis, these results are consistent with the absence of PCM1 causing a delay in centriole biogenesis.

In addition, basal bodies of *Pcm1*⁻/⁻ ependymal cells displayed disrupted translational polarity of basal bodies within the apical domain, which persisted until P16 (*Figure 3B, G, H*, *Figure 3—figure supplement 1A, B*). Basal body positioning within the apical domain is thought to be independent of ciliary motility, suggesting roles for PCM1 in ependymal cells beyond motility (*Kishimoto and Sawamoto, 2012*; *Mirzadeh et al., 2010b*).

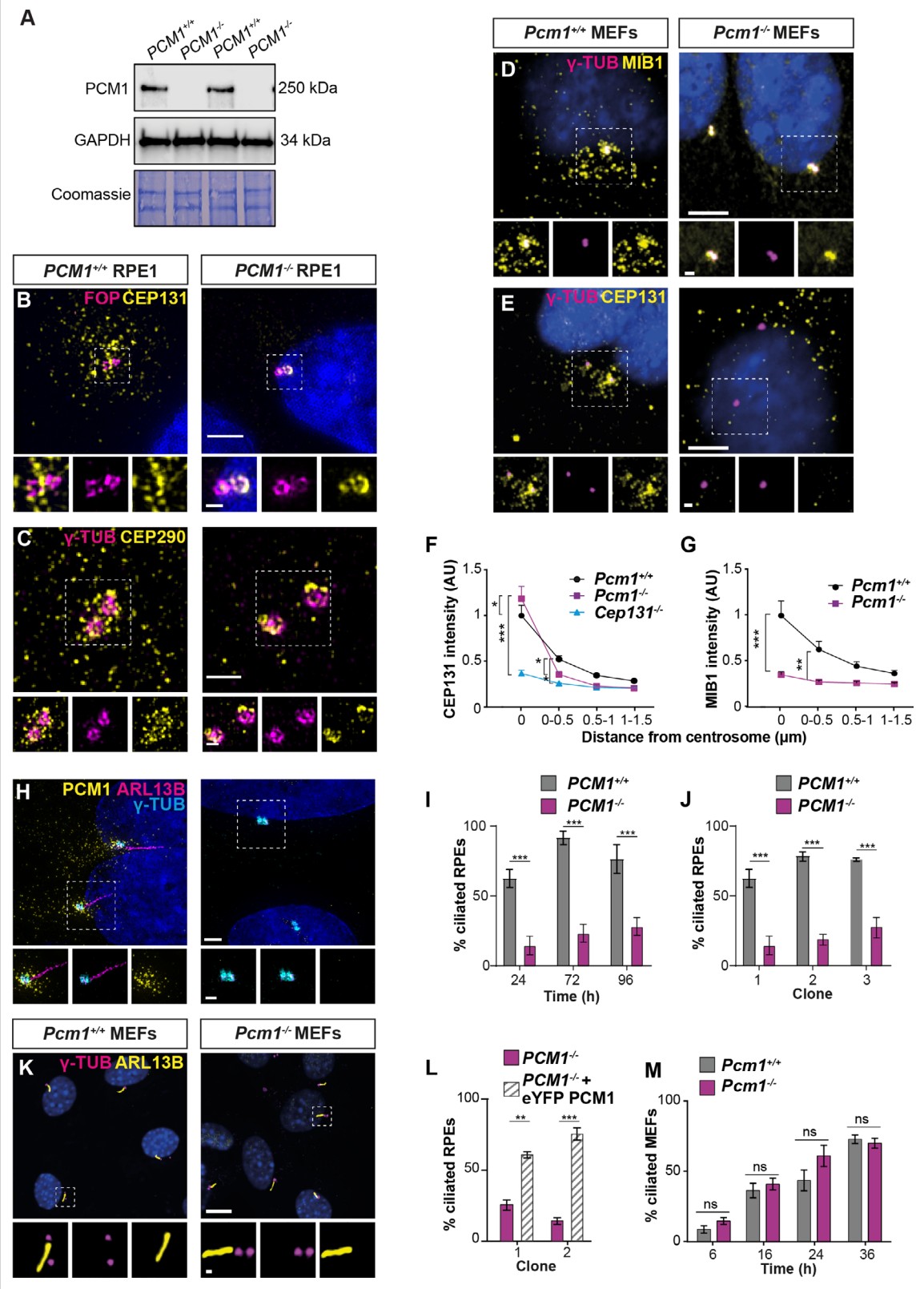

**Figure 4.** PCM1 is essential for centriolar satellite integrity and, in some cell types, ciliogenesis. (**A**) Immunoblot of wild-type and *PCM1⁻/⁻* retinal pigmented epithelial 1 (RPE1) cell lysates for PCM1 and GAPDH (loading control). Gel stained with Coomassie blue. (**B**) Wild-type and *PCM1⁻/⁻* RPE1 cells immunostained for CEP131 (yellow), centrioles (FOP, magenta), and nuclei (DAPI, blue). (**C**) Wild-type and *PCM1⁻/⁻* RPE1 cells immunostained for CEP290 (yellow), centrioles (γ-TUB, magenta), and nuclei (DAPI, blue). (**D, E**) Wild-type and *Pcm1⁻/⁻* mouse embryonic fibroblasts (MEFs) immunostained

*Figure 4 continued*

for centrioles (γ-TUB, magenta), and nuclei (DAPI, blue) with CEP131 (**D**) or MIB1 (**E**) (yellow). (**F**) CEP131 intensity as a function of distance from the centrosome. *Cep131⁻ᐟ⁻* MEFs are included as a control (***Hall et al., 2013***). Two-way analysis of variance (ANOVA), comparing wild-type to mutants, with Dunnett correction for multiple testing: $*p < 0.05$, $**p < 0.01$, $***p < 0.001$. Error bars represent standard error of the mean (SEM). (**G**) MIB1 intensity as a function of distance from the centrosome. (**H**) Wild-type and *PCM1⁻ᐟ⁻* RPE1 cells immunostained for PCM1 (yellow), cilia (ARL13B, magenta), centrioles (γ-TUB, cyan), and nuclei (DAPI, blue). (**I**) Percentage of wild-type and *PCM1⁻ᐟ⁻* RPE1 cells serum starved for 24, 72, or 96 hr that are ciliated. Bar graphs show means ± standard deviation (SD). Unpaired Student's *t*-test: $***p < 0.001$. $n > 100$ cells from 3 replicates. (**J**) Percentage of three control (treated with non-targeting sgRNA) and *PCM1⁻ᐟ⁻* RPE1 clonal lines, serum starved for 24 hr that are ciliated. Bar graphs show means ± SEM. Unpaired Student's *t*-test: $***p < 0.001$. $n > 100$ cells from 2 replicates. (**K**) Wild-type and *Pcm1⁻ᐟ⁻* MEFs immunostained for cilia (ARL13B, yellow), centrioles (γ-TUB, magenta), and nuclei (DAPI, blue). (**L**) Percentage of two *PCM1⁻ᐟ⁻* RPE1 clonal lines with and without eYFP-PCM1 expression serum starved for 24 hr. Bar graphs show means ± SEM. Unpaired Student's *t*-test: $**p < 0.01$, $***p < 0.001$. $n > 100$ cells from 2 replicates. (**M**) Percentage of wild-type and *Pcm1⁻ᐟ⁻* MEFs serum starved for 6–36 hr that are ciliated. Bar graphs show means ± SEM. $n = 3$ MEF lines from different embryos per genotype. Student's *t*-test, ns: not significant. Scale bars: 2 μm (**B**), 1 μm (**C**), 0.5 μm (**B, C** insets), 5 μm (**D, E**), 1 μm (**D, E** insets), 10 μm (**H, K**), and 1 μm (**H, K** insets).

The online version of this article includes the following video, source data, and figure supplement(s) for figure 4:

**Source data 1.** Full uncropped immunoblots for *Figure 4G*, labeled and unlabeled.

**Figure supplement 1.** PCM1 is dispensable for ciliogenesis in mouse embryonic fibroblasts (MEFs).

**Figure 4—video 1.** Centriolar satellites frequently fuse and divide near the basal body.

https://elifesciences.org/articles/79299/figures#fig4video1

Interestingly, *Pcm1⁻ᐟ⁻* ependymal cells contained highly elongated FOP- and Centrin-containing centriole-like structures measuring 5.0 ± 1.9 μm (mean ± standard deviation [SD]) in length (***Figure 3—figure supplement 2A–F***). Together these results suggest disrupted centriole biogenesis and migration in the absence of PCM1.

At P5, there were fewer *Pcm1⁻ᐟ⁻* ciliated ependymal cells. However, by P16, the number of ciliated *Pcm1⁻ᐟ⁻* ependymal cells was equivalent to control ventricles (***Figure 3A, B, D***, ***Figure 3—figure supplement 1A, B***). This delay in ependymal ciliogenesis in the absence of PCM1 could be secondary to the delay in centriole biogenesis. At P3, *Pcm1⁻ᐟ⁻* ependymal cilia displayed ultrastructural defects, including missing microtubule doublets and fused axonemes (***Figure 3I, K***).

To further analyze the function of PCM1 in multiciliogenesis, we cultured primary ependymal cells (***Guirao et al., 2010***) isolated from P0–P3 wild-type control and *Pcm1⁻ᐟ⁻* mice. These *Pcm1⁻ᐟ⁻* ependymal cells possessed fewer centrioles at the disengagement stage of centriole biogenesis, but once the cells became multiciliated had normal numbers of centrioles (***Figure 3—figure supplement 1C, E, G***). In culture, *Pcm1⁻ᐟ⁻* ependymal cells formed fewer cilia than control ependymal cells (***Figure 3J, L***, ***Figure 3—figure supplement 1C, F***). High-speed video microscopy revealed that *Pcm1⁻ᐟ⁻* ependymal cilia beat slowly and uncoordinatedly (***Figure 3J, M, N***, ***Figure 3—videos 1–3***). These findings further support the conclusion that the lack of PCM1 causes a delay in centriole biogenesis and disrupts motile ciliary function.

Thus, PCM1 is not essential for ciliogenesis, but is required for timely basal body biogenesis, maturation, migration, and ciliogenesis in ependymal cells. We propose that hydrocephaly in *Pcm1⁻ᐟ⁻* mice is caused by delayed ependymal cell ciliogenesis and compromised ciliary motility.

Like the brain ventricles, the trachea is lined by motile multiciliated cells. To examine whether PCM1 also promotes ciliogenesis and ciliary motility in the airways, we examined mouse tracheal basal bodies and cilia by immunofluorescence. *Pcm1⁻ᐟ⁻* tracheal multiciliated cells in vivo did not display decreased numbers of basal bodies or cilia at P5, or altered axonemal ultrastructure at 6 months of age (***Figure 3—figure supplement 3B–D***). High-speed video microscopy revealed *Pcm1⁻ᐟ⁻* tracheal cilia beat at normal frequency (***Figure 3—figure supplement 3E***, ***Figure 3—videos 4 and 5***).

To investigate the dynamics of ciliogenesis in these cells, we differentiated mouse tracheal epithelial cells (mTECs) into multiciliated cells in vitro (***Eenjes et al., 2018***; ***You et al., 2002***). Concurring with a previous reports on the dispensability of PCM1 in mTECs (***Vladar and Stearns, 2007***), *Pcm1⁻ᐟ⁻* mTECs displayed normal basal body biogenesis, ciliogenesis, and ciliary beat frequency (***Figure 3—figure supplement 3A, F***). However, proteomic analysis of differentiating *Pcm1⁻ᐟ⁻* mTECs revealed that many motile ciliary proteins, including dynein motors, dynein assembly factors and dynein docking factors, were reduced early in ciliogenesis (air–liquid interface [ALI] day 7) (***Figure 3—figure supplement 3G***, ***Supplementary file 5***). Similar to the transitory delay we observed in *Pcm1⁻ᐟ⁻* ependymal cell ciliogenesis, proteomic differences in *Pcm1⁻ᐟ⁻* mTECs resolved by ALI day 21 (***Figure 3—figure***

*supplement 3G*, *Supplementary file 5*). Thus, as in ependymal cells, PCM1 promotes timely cilia maturation in tracheal cells.

In multiciliated cells, PCM1 and other centriolar satellite proteins including CEP131 and PCNT localize to fibrogranular material, satellite-like networks (*Zhao et al., 2021*). Consistent with previous findings from Zhao et al., we found that CEP131 in mTECs lacking PCM1 localized not to fibrogranular material but to centrioles, (*Figure 3O*). Similarly, in *Pcm1⁻/⁻* ependymal cells, CEP131 mislocalized to the centrioles, although rather than being absent from the fibrogranular material, this non-centriolar CEP131 pool became more elongated (*Figure 3—figure supplement 1H*). Not all centriolar satellite components behaved similarly in the absence of PCM1; localization of PCNT was normal in *Pcm1⁻/⁻* ependymal cells (*Figure 3—figure supplement 1I*). Thus, fibrogranular material in the absence of PCM1 can either be disrupted or change its distribution in different multiciliated cell types. Together, these results suggest that PCM1 is required for fibrogranular material integrity, centriole biogenesis, and migration, and timely ciliogenesis in multiciliated cells.

## PCM1 is required for centriolar satellite integrity

To assess whether PCM1 is essential for centriolar satellite integrity, we analyzed *Pcm1⁻/⁻* MEFs and *PCM1⁻/⁻* RPE1 cells (*Kumar et al., 2021*). Immunoblot and immunofluorescence analyses confirmed loss of PCM1 protein in the mutant cells (*Figure 1A, B*, *Figure 4A, H*). In addition to PCM1 and CEP131, centriolar satellites contain proteins such as CEP290 and the E3 ligase MIB1 (*Hall et al., 2013*; *Staples et al., 2012*; *Villumsen et al., 2013*). In control RPE1 cells, CEP131 and CEP290 localized to both centriolar satellites and to the centrioles themselves. In *PCM1⁻/⁻* RPE1 cells, the centriolar satellite pool of CEP131 was absent, CEP290 was reduced and dispersed, and both displayed increased accumulation at centrioles (*Figure 4B, C*). In control MEFs, CEP131 and MIB1 localized to both centriolar satellites and to the centrioles themselves. In *Pcm1⁻/⁻* MEFs, the centriolar satellite pool of CEP131 was absent and MIB1 was reduced and dispersed, with CEP131 displaying increased accumulation at centrioles, similar to *Pcm1⁻/⁻* tracheal epithelial cells (*Figure 4D–G*). We conclude that PCM1 is critical for centriolar satellite integrity. In the absence of satellites, some satellite proteins (e.g., CEP131 and CEP290) over-accumulate at centrioles, while others (e.g., MIB1) do not, highlighting the protein-specific role centriolar satellites play in controlling centriolar localization. We propose that centriolar satellites both deliver and remove select cargos from centrioles.

One way in which satellites could traffic cargos to and from centrioles would be via their movement within the cell. To visualize PCM1, we engineered mice expressing a fusion of PCM1 and the SNAP tag (*Keppler et al., 2003*) from the *Pcm1* locus. We derived MEFs from *Pcm1*ˢᴺᴬᴾ mice, covalently labeled PCM1-SNAP with tetramethylrhodamine (*Crivat and Taraska, 2012*), and imaged centriolar satellite movement relative to cilia. Consistent with previous reports (*Conkar et al., 2019*), centriolar satellites moved both toward and away from the ciliary base, with frequent fission and fusion at the ciliary base (*Figure 4—video 1*).

To further explore how centriolar satellites promote ciliogenesis, we examined ciliogenesis in MEFs and RPE1 cells lacking PCM1. In accordance with previous observations (*Odabasi et al., 2019*; *Wang et al., 2016*), ciliogenesis was abrogated in several *PCM1⁻/⁻* RPE1 cell lines (*Figure 4H, J*) and could be rescued by expression of eYFP-PCM1 (*Figure 4L*).

In marked contrast, and consistent with the tissue-specific effects of loss of PCM1 on ciliogenesis (*Figure 1—figure supplement 3*), ciliogenesis was not perturbed in *Pcm1⁻/⁻* MEFs, with *Pcm1⁻/⁻* MEFs displaying cilia number, centrosome number and cilia length indistinguishable from those of controls (*Figure 4K, M*, *Figure 4—figure supplement 1A–D*). Thus, PCM1, despite broad roles in regulating the centriolar localization of proteins such as CEP131, plays cell-type-specific roles in ciliogenesis.

## PCM1 is dispensable for removal of Centrobin and assembly of distal and subdistal appendages

An early step in ciliogenesis is the removal of daughter centriole-specific protein Centrobin (*Stephen et al., 2015*; *Wang et al., 2018*). A previous study proposed a role for PCM1-localizing centriolar satellites in regulating the abundance of Talpid3, a component of the distal centriole implicated in the removal of Centrobin from the mother centriole (*Wang et al., 2018*; *Wang et al., 2016*). We found that both Talpid3 and Centrobin localization to centrioles in *PCM1⁻/⁻* RPE1 cells was equivalent to those of controls (*Figure 5—figure supplement 1A–E*). Thus, Talpid3 recruitment to centrioles and

Centrobin removal from the mother centriole are not dependent upon PCM1 or, by extension, centriolar satellites.

Distal appendages anchor the mother centriole to the ciliary membrane and subdistal appendages position the cilium within cells (*Mazo et al., 2016*; *Schmidt et al., 2012*; *Sillibourne et al., 2013*; *Tanos et al., 2013*). Since centriolar satellite cargos (e.g., CEP90, OFD1, and MNR) are essential for ciliogenesis and distal appendage assembly (*Kumar et al., 2021*), we hypothesized that PCM1 may participate in distal or subdistal appendage formation. To test this hypothesis, we examined localization of components of the distal (i.e., FBF1 and ANKRD26) and subdistal appendages (i.e., Ninein) at the mother centriole. In *PCM1$^{-/-}$* RPE1 cells, both distal and subdistal appendage components localized to the mother centriole (*Figure 5—figure supplement 1F–K*), although the amount of distal appendage proteins at the mother centriole was slightly reduced. Serial section transmission electron microscopy (TEM) confirmed that subdistal and distal appendages were present in *PCM1$^{-/-}$* RPE1 cells (*Figure 5—figure supplement 2*). Therefore, centriolar satellites are not required for the assembly of distal or subdistal appendages at the mother centriole.

## PCM1 promotes formation of the ciliary vesicle

After acquiring distal appendages, the mother centriole docks to preciliary vesicles, small vesicles which accumulate at the distal appendages of the mother centriole and are converted into a larger ciliary vesicle (*Schmidt et al., 2012*; *Sillibourne et al., 2013*; *Tanos et al., 2013*). To further examine the cause of reduced ciliogenesis in RPE1 cells lacking centriolar satellites, we investigated whether preciliary vesicle docking or ciliary vesicle formation depends on PCM1.

Myosin-Va adorns preciliary and ciliary vesicles (*Wu et al., 2018*). Using 3D-SIM imaging of Myosin-Va, we identified preciliary vesicles at the basal bodies of control RPE1 cells soon after the induction of ciliogenesis (i.e., after 1 hr of serum starvation). *PCM1$^{-/-}$* RPE1 cells showed reduced Myosin-Va at preciliary vesicles (*Figure 5A, B*), suggesting that centriolar satellites promote timely centriolar docking of preciliary vesicles.

Since Myosin-Va marks both preciliary and ciliary vesicles, we more specifically assessed ciliary vesicle formation at the mother centriole by examining the localization of RAB34. RAB34 is a GTPase that marks the ciliary vesicle early in ciliogenesis and, later, the ciliary sheath (*Ganga et al., 2021*). Using 3D-SIM imaging, we observed RAB34 at the centrosome of wild-type RPE1 cells after 1-hr serum starvation, and at both centrosomes and ciliary sheaths after 24-hr serum starvation (*Figure 5C, D*). *PCM1$^{-/-}$* RPE1 cells showed reduced RAB34 at both centrosomes and ciliary sheaths (*Figure 5C, D*), suggesting that centriolar satellites promote timely docking of the mother centriole to the preciliary vesicles, and the fusion of preciliary vesicles into a ciliary vesicle.

To assess ciliary vesicle formation using a complementary approach, we performed serial section TEM of control and *PCM1$^{-/-}$* RPE1 cells early in ciliogenesis (i.e., after 1 hr of serum starvation). We quantified preciliary and ciliary vesicles at mother centrioles. In *PCM1$^{-/-}$* cells, mother centrioles (identified by the presence of distal and subdistal appendages) exhibited reduced association with preciliary and ciliary vesicles (*Figure 5E, F*, *Figure 5—figure supplement 2*). Thus, centriolar satellites promote the attachment of the mother centriole to preciliary vesicles and formation of the ciliary vesicle, important early steps in ciliogenesis.

## PCM1 promotes CP110 and CEP97 removal from the mother centriole

In vertebrates, CP110 is required for docking of the mother centriole to preciliary vesicles (*Walentek et al., 2016*; *Yadav et al., 2016*) and is removed from the mother centriole subsequent to formation of the ciliary vesicle (*Lu et al., 2015*; *Wu et al., 2018*). The cap comprised of CP110 and CEP97 inhibits ciliogenesis, and its removal from the distal mother centriole is important for axoneme elongation (*Spektor et al., 2007*; *Yadav et al., 2016*). Since PCM1 promotes timely ciliary vesicle formation, we examined whether CP110 and CEP97 removal also depends on PCM1. In contrast to control cells, CP110 and CEP97 persisted at the distal mother centriole in *PCM1$^{-/-}$* RPE1 cells after 24 hr of serum starvation (*Figure 6A–D*).

Interestingly, in wild-type MEFs, a small amount of CP110 persisted on the mother centriole even after axoneme formation (*Figure 6E, G, H*). Strikingly, mother centrioles in *Pcm1$^{-/-}$* MEFs had CP110 levels comparable to daughter centrioles after 24 hr serum starvation, despite undergoing

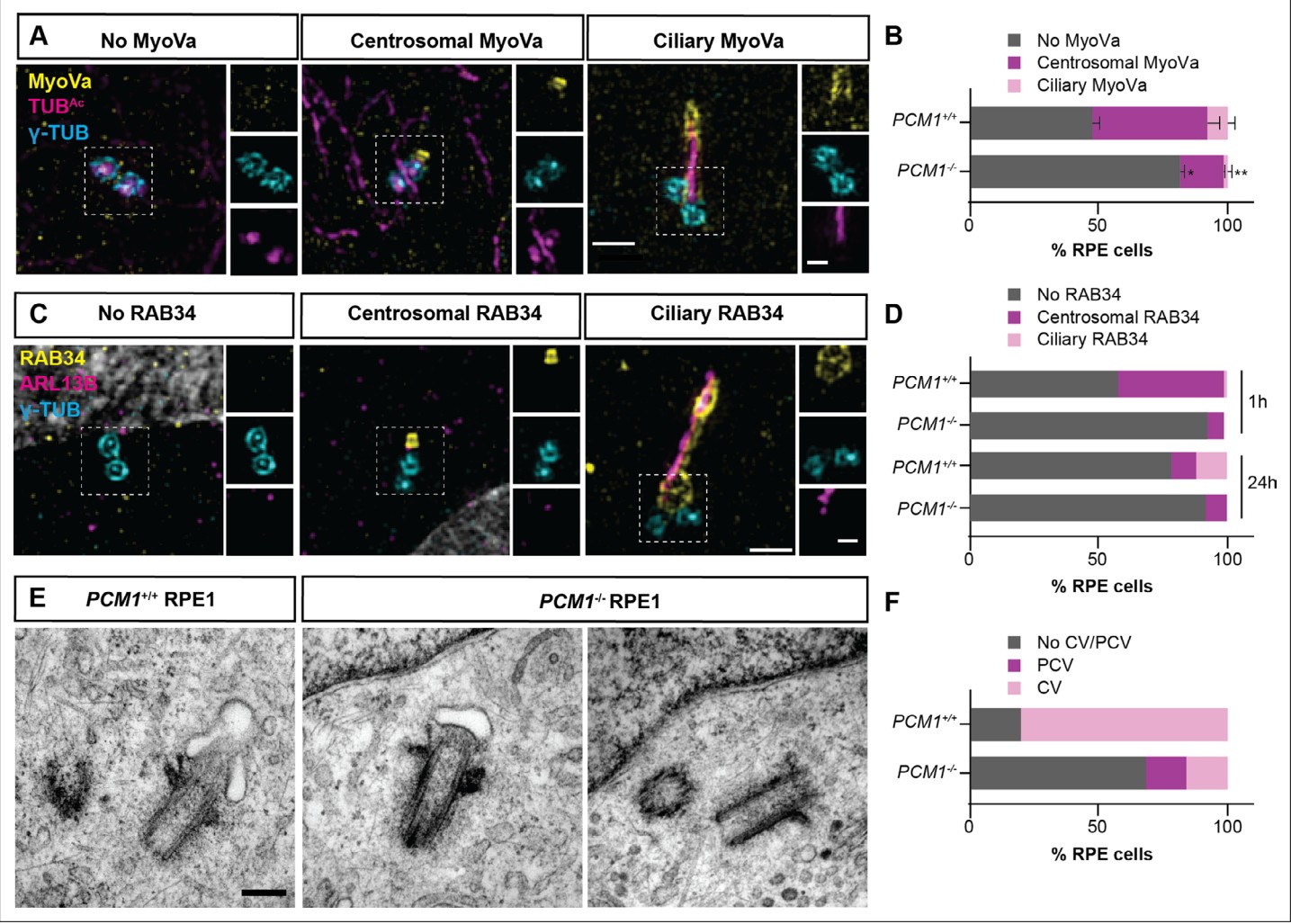

**Figure 5.** PCM1 promotes mother centriole docking to preciliary vesicles. (**A**) 3D-SIM images of Myosin-Va (MyoVa, yellow), centrioles (γ-TUB, cyan) and cilia (TUB$^{Ac}$, magenta) in wild-type and $PCM1^{-/-}$ RPE cells 1 hr after serum starvation. Scale bars: 1 and 0.5 μm for main panels and insets, respectively. (**B**) Percentage of wild-type and $PCM1^{-/-}$ retinal pigmented epithelial 1 (RPE1) cells with no MyoVa at centrosomes, MyoVa at centrosomes, and MyoVa at cilia. Bar graphs show means ± standard error of the mean (SEM). Unpaired Student's $t$-test compared with wild-type: *$p < 0.05$, **$p < 0.005$. $n > 50$ cells from 2 replicates. (**C**) 3D-SIM images of RPE1 cells immunostained with RAB34 (yellow), centrioles (γ-TUB, cyan), and cilia (ARL13B, magenta). Scale bars: 1 and 0.5 μm for main panels and insets, respectively. (**D**) Percentage of wild-type and $PCM1^{-/-}$ RPE cells 1 and 24 hr after serum starvation exhibiting no centrosomal RAB34, RAB34 at centrosomes, and RAB34 at cilia. $n > 100$ cells. (**E**) Serial-section transmission electron microscopy (TEM) of RPE1 cells during early ciliogenesis (1 hr after serum starvation). Scale bar: 200 nm. (**F**) Percentage of wild-type and $PCM1^{-/-}$ RPE1 cells in which TEM images demonstrate basal body association with preciliary vesicles (PCV) or ciliary vesicles (CV). $n = 5$–20 cells.

The online version of this article includes the following figure supplement(s) for figure 5:

**Figure supplement 1.** PCM1 is dispensable for mother centriole maturation.

**Figure supplement 2.** PCM1 promotes mother centriole association with vesicles.

ciliogenesis at rates equal to that of wild-type cells (***Figure 6E, G, H***). Thus, PCM1 is essential for removing CP110 from the mother centriole, but CP110 removal is not required for ciliogenesis in MEFs.

Similar to RPE1 cells and MEFs, in $Pcm1^{-/-}$ ependymal cells in vivo, CP110 levels were elevated at P3, an age when ependymal calls are engaged in ciliogenesis (***Figure 6F, I***). Moreover, CP110 levels were elevated at the multiple basal bodies of $Pcm1^{-/-}$ tracheal multiciliated cells (***Figure 6—figure supplement 1***). Thus, diverse cell types require PCM1 to remove CP110 from the mother centriole, despite differentially requiring PCM1 for ciliogenesis.

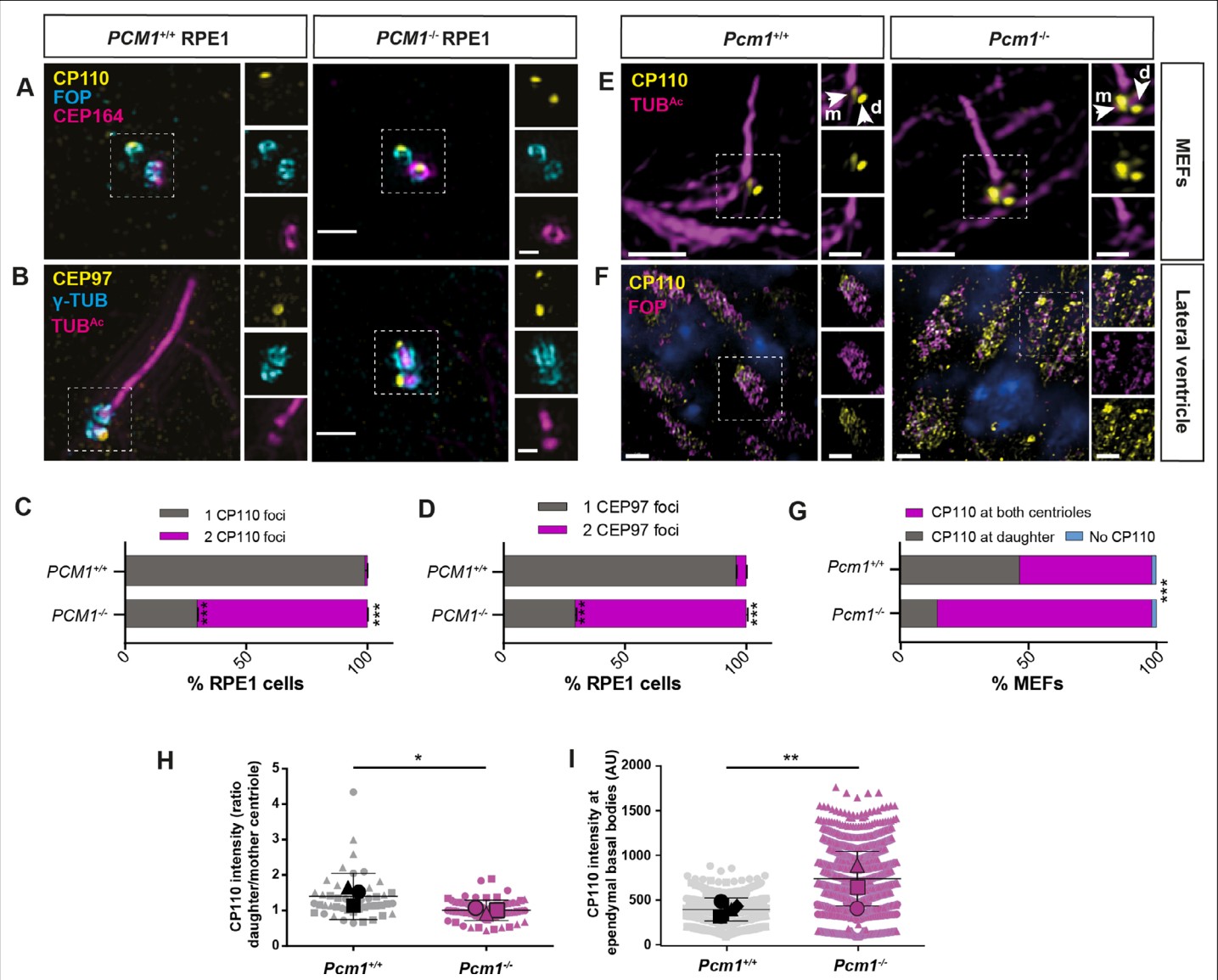

**Figure 6.** PCM1 promotes removal of CP110 and CEP97 from the mother centriole. (**A**) Wild-type and *PCM1−/−* RPE1 cells serum starved for 24 hr immunostained for CP110 (yellow), centrioles (FOP, cyan), and distal appendages (CEP164, magenta). (**B**) Wild-type and *PCM1−/−* RPE1 cells serum starved for 24 hr immunostained for CEP97 (yellow), centrioles (γ-TUB, cyan), and cilia (TUB^Ac, magenta). (**C**) Percentage of wild-type and *PCM1−/−* RPE1 cells with CP110 levels at one or two centrioles. Bar graphs show means ± standard error of the mean (SEM). Unpaired Student's *t*-test compared with wild-type: ***p < 0.0005. *n* > 50 cells from 2 replicates. (**D**) Percentage of wild-type and *PCM1−/−* RPE1 cells with CEP97 levels at one or two centrioles. Bar graphs show means ± SEM. Unpaired Student's *t*-test compared with wild-type: ***p < 0.0005. *n* > 50 cells from 2 replicates. (**E**) Wild-type and *Pcm1−/−* MEFs serum starved for 24 hr and immunostained for CP110 (yellow) and cilia (TUB^Ac, magenta). (**F**) Wild-type and *Pcm1−/−* lateral ventricular wall immunostained for CP110 (yellow), basal bodies (FOP, cyan), and nuclei (DAPI, blue). (**G**) Percentage of wild-type and *Pcm1−/−* MEFs serum starved for 24 hr with CP110 levels at none, one or two centrioles. Chi squared test ***p < 0.001. (**H**) The ratio of CP110 intensity on daughter and mother centrioles in wild-type and *Pcm1−/−* MEFs serum starved for 24 hr. (**I**) Intensity of CP110 in wild-type and *Pcm1−/−* ependymal cells. *n* = 3 per genotype. Large symbols represent individual animals, small symbols represent individual cells. Student's *t*-test, *p < 0.05, **p < 0.01, ***p < 0.001. Scale bars represent 1 μm (main panel) and 0.5 μm (inset) (**A, B**), represent 5 μm (main panel) and 1 μm (inset) (**E**), and 2 μm (**F**).

The online version of this article includes the following figure supplement(s) for figure 6:

**Figure supplement 1.** PCM1 promotes removal of CP110 from basal bodies of airway multiciliated cells.

## PCM1 promotes transition zone formation and intraflagellar transport (IFT) recruitment

Following ciliary vesicle docking and removal of CP110 and CEP97 from the mother centriole, ciliogenesis proceeds with transition zone construction and IFT recruitment (*Ishikawa and Marshall, 2011*). Since PCM1 promotes ciliary vesicle docking and CP110 and CEP97 removal, we hypothesized that in cells lacking PCM1 the subsequent engagement of IFT and transition zone components would be compromised.

To test this hypothesis, we immunostained control and *PCM1⁻/⁻* RPE1 cells with antibodies to IFT88 and IFT81. As expected, IFT88 and IFT81 localized to mother centrioles and along the length of cilia in control cells (*Figure 7A, C*). Localization of both IFT88 and IFT81 at mother centrioles was reduced in *PCM1⁻/⁻* RPE1 cells (*Figure 7A–D*), suggesting that IFT recruitment to the mother centriole is promoted by centriolar satellites. In contrast, ciliary and basal body levels of IFT88 were normal in *Pcm1⁻/⁻* MEFs (*Figure 7—figure supplement 1A, B*), indicating a concordance between PCM1-dependent IFT recruitment and ciliogenesis.

The transition zone controls ciliary protein composition. We determined whether PCM1 was required for the formation of the transition zone by assessing the localization of CEP162, an axoneme-associated protein that recruits components of the transition zone, such as RPGRIP1L (*Wang et al., 2013b*). Recruitment of CEP162 to the mother centriole was unaffected in *PCM1⁻/⁻* RPE1 cells (*Figure 7E, F*). In contrast, *PCM1⁻/⁻* RPE1 cells exhibited reduced RPGRIP1L at the transition zone (*Figure 7G, H*). Therefore, centriolar satellites promote both IFT recruitment and transition zone formation at RPE1 cell mother centrioles.

## Centriolar satellites restrict CP110 and CEP97 levels at centrioles

To explore the mechanisms by which centriolar satellites regulate CP110 and CEP97 levels at the centrioles, we examined the localization of TTBK2. TTBK2 is a kinase recruited by CEP164, a distal appendage component required to remove CP110 and CEP97 from mother centrioles (*Goetz et al., 2012*). In *PCM1⁻/⁻* RPE1 cells, TTBK2 recruitment to distal mother centrioles was equivalent to that of control cells (*Figure 7I, J*). These results suggest that centriolar satellites regulate CP110 and CEP97 removal from the distal mother centriole through a mechanism independent of TTBK2 recruitment.

As PCM1 is dispensable for the localization of TTBK2 at the distal mother centriole, we considered alternative mechanisms by which PCM1 may regulate local CP110 and CEP97 levels at the mother centriole. Since centriolar satellites are highly dynamic and localization of CP110 and CEP97 is actively controlled at the initiation of ciliogenesis, we hypothesized that CP110 and CEP97 are transported away from the centrioles via satellites. A prediction of this model is that CP110 and CEP97 should localize to satellites.

We examined RPE1 cells for CP110 and CEP97 and found that, indeed, CP110 and CEP97 colocalized with PCM1 and CEP290 at centriolar satellites in cycling cells (*Figure 8A, B*, *Figure 8—figure supplement 1A, C*). Moreover, this satellite pool of CP110 was absent in *PCM1⁻/⁻* RPE cells (*Figure 8—figure supplement 1B*). Consistent with CP110 and CEP97 co-localizing with PCM1 at centriolar satellites, CP110 and CEP97 co-immunoprecipitated with PCM1 in cycling cells (*Figure 8C*).

By examining RPE1 cells at different timepoints after serum depletion, we observed that the localization of CP110 and CEP97 to centrioles and centriolar satellites was dynamic: 1 hr after initiating ciliogenesis, CP110 and CEP97 at satellites decreased and, by 24 hr of serum depletion, CP110 and CEP97 were absent from the mother centriole (*Figure 8A, B*).

CP110 interacts with satellite protein CEP290 (*Tsang et al., 2008*), so we hypothesized that CEP290 may hold CP110 at the satellites. Consistent with this model, CP110 no longer localized to satellites in cycling RPE1 cells upon *CEP290* knockdown (*Figure 8—figure supplement 1D*). We propose that CP110 and CEP97 are centriolar satellite cargos which are wicked away from mother centrioles by centriolar satellites during early ciliogenesis.

If centriolar satellites transport CP110 and CEP97 away from centrioles as an early step in ciliogenesis, PCM1 may be critical for CP110 turnover. We therefore assessed total CP110 and CEP290 protein levels by immunoblot. In serum-starved *PCM1⁻/⁻* RPE1 cells, both CEP290 and CP110 were modestly elevated relative to serum-starved control cells (*Figure 8D, E*). Similarly, in synchronized RPE1 cells, CP110 levels were increased in the absence of PCM1, most markedly during mitosis and G0 (*Figure 8—figure supplement 1E*).

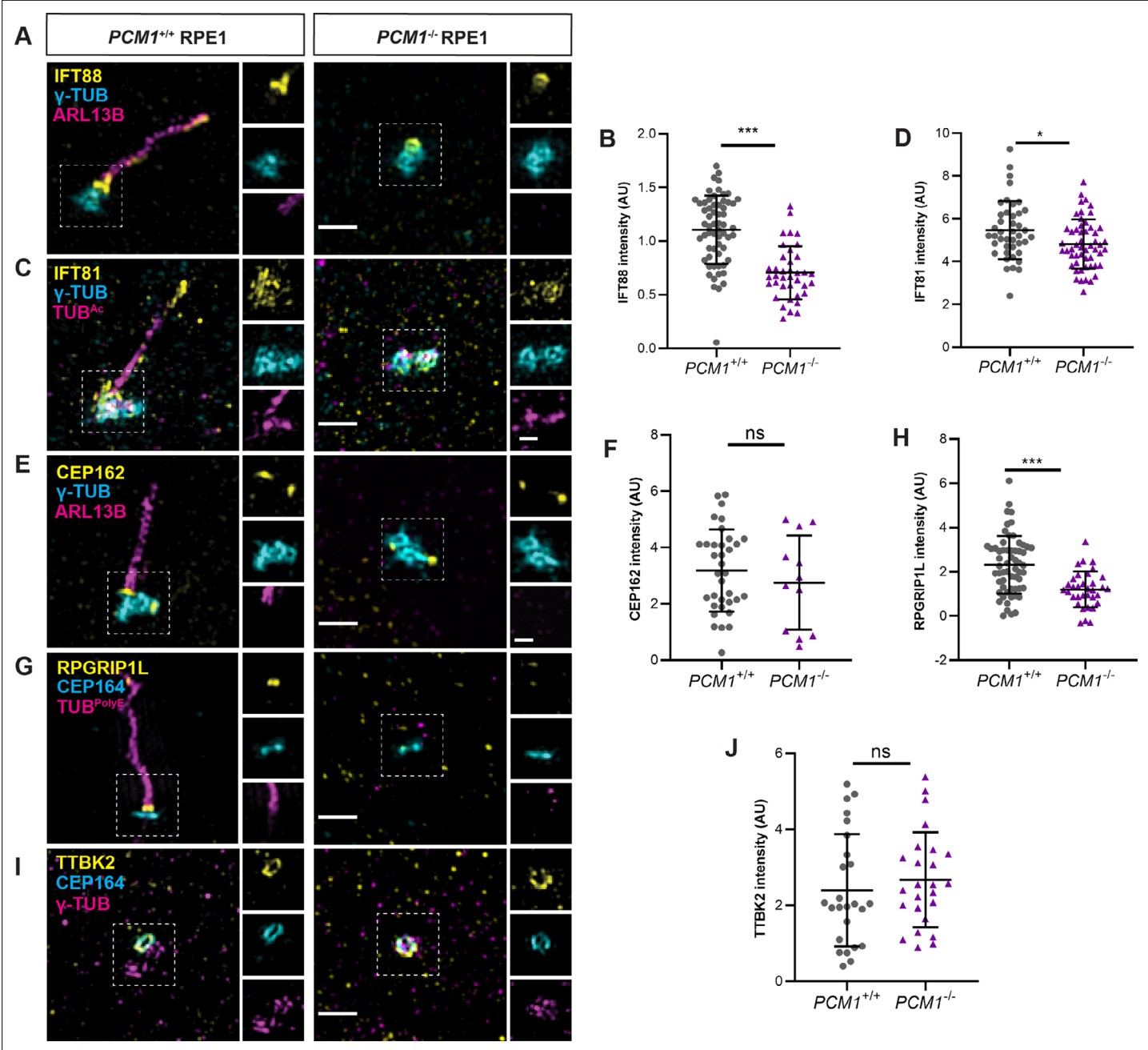

**Figure 7.** PCM1 promotes IFT recruitment and transition zone formation. (**A**) Wild-type and *PCM1⁻/⁻* RPE1 cells immunostained for IFT88 (yellow), centrioles (γ-TUB, cyan), and cilia (ARL13B, magenta). (**B**) Quantification of IFT88 intensity at basal bodies. (**C**) Immunostaining for IFT81 (yellow), centrioles (γ-TUB, cyan), and cilia (TUB^Ac, magenta). (**D**) Quantification of IFT81 intensity at basal bodies. (**E**) Immunostaining for CEP162 (yellow), centrioles (γ-TUB, cyan), and cilia (ARL13B, magenta). (**F**) Quantification of CEP162 intensity at basal bodies. (**G**) Immunostaining for transition zone component RPGRIP1L (yellow), distal appendages (CEP164, cyan), and cilia (TUB^polyE, magenta). (**H**) Quantification of RPGRIP1L intensity at transition zones. (**I**) Immunostaining for TTBK2 (yellow), distal appendages (CEP164, cyan), and centrioles (γ-TUB, magenta). (**J**) Quantification of TTBK2 intensity at basal bodies. Scale bars in main figures represent 1 μm and in insets represent 0.5 μm. Bar graphs show means ± standard deviation (SD) from 2 experiments. Student's *t*-test: *p < 0.05, ***p < 0.001, ns, not significant.

The online version of this article includes the following figure supplement(s) for figure 7:

**Figure supplement 1.** PCM1 does not control IFT88 levels in MEF cilia.

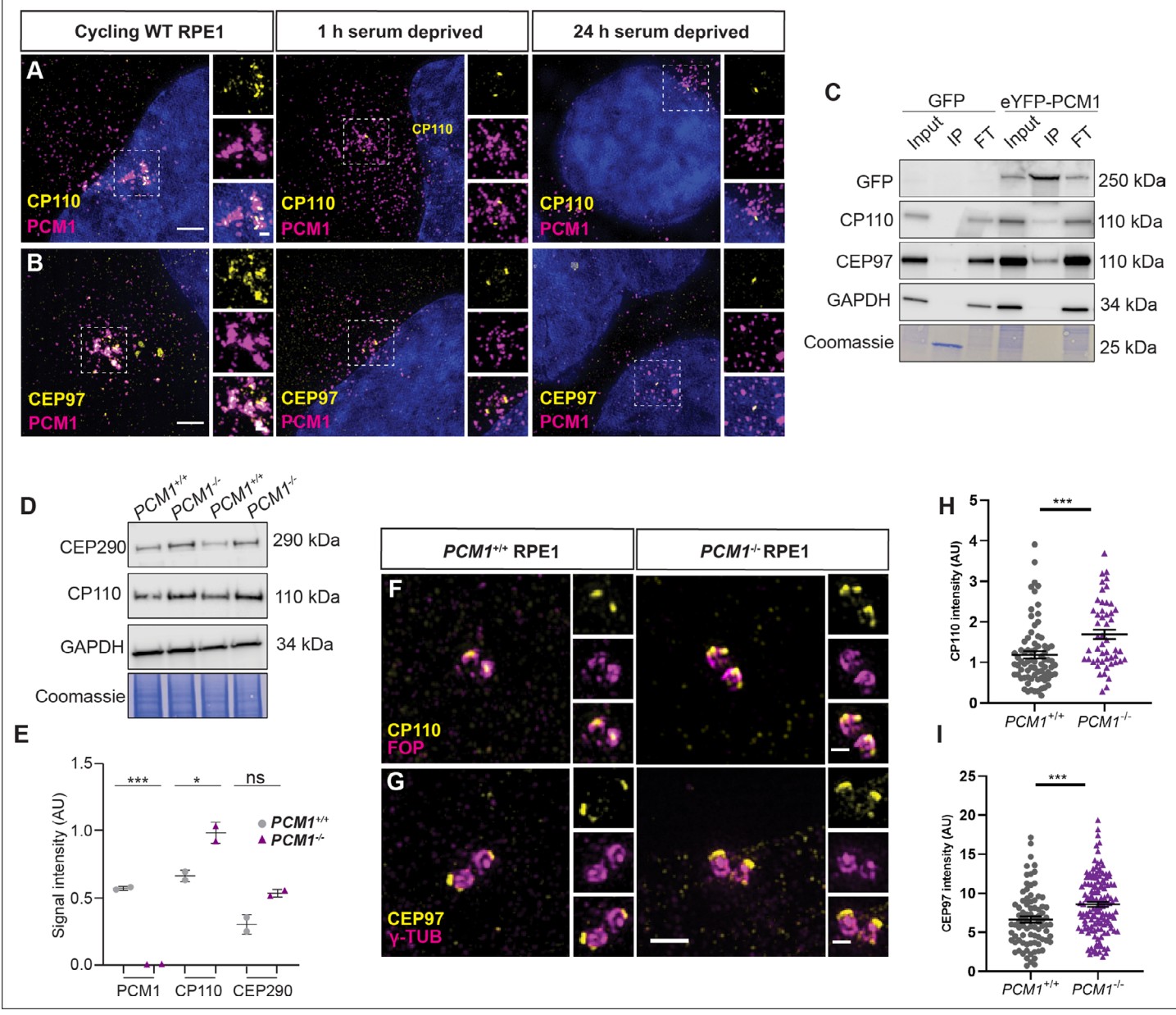

**Figure 8.** PCM1 restricts CP110 and CEP97 localization to distal mother centrioles. (**A**) Wild-type and *PCM1⁻/⁻* RPE1 cells immunostained for CP110 (yellow), centriolar satellites (PCM1, magenta), and nuclei (DAPI, blue) in cells with serum (cycling) or 1 or 24 hr after withdrawing serum. (**B**) Immunostaining for CEP97 (yellow), centriolar satellites (PCM1, magenta), and nuclei (DAPI, blue). (**C**) Total cell lysates of *PCM1⁻/⁻* RPE1 cell lines stably expressing eGFP or eYFP-PCM1 subjected to immunoprecipitation with anti-GFP. Precipitating proteins were immunoblotted for GFP, CP110, CEP97, and GAPDH. IP: eluate. FT: flow through. (**D**) Immunoblot of wild-type and *PCM1⁻/⁻* RPE1 cell lines lysates for CP110 and GAPDH, as well as Coomassie stain of gels. Cells were deprived of serum for 24 hr prior to lysis. (**E**) Quantification of PCM1 and CP110 levels from immunoblots. Bar graphs show means ± SEM from 2 experiments. (**F**) Wild-type and *PCM1⁻/⁻* RPE1 cells immunostained for CP110 (yellow) and centrioles (FOP, magenta). Cycling cells were treated with nocodazole to disperse the centriolar satellite pool of CP110, leaving the centriolar pool. (**G**) Immunostaining for CEP97 (yellow) and centrioles (γ-TUB, magenta) in cycling cells treated with nocodazole. (**H**) Quantification of CP110 levels at centrioles stained as in F. (**I**) Quantification of CEP97 levels at centrioles stained as in G. Scale bars: 1 and 0.5 μm in main panels and insets, respectively. Bar graphs show means ± SEM and n>30 cells from 2 experiments. Student's *t*-test: *p < 0.05, ***p < 0.001, ns, not significant.

The online version of this article includes the following source data and figure supplement(s) for figure 8:

**Source data 1.** Full uncropped immunoblots for *Figure 8C, I* and *Figure 8—figure supplement 1E*, labeled and unlabeled.

**Figure supplement 1.** CP110 localizes to satellites in a CEP290-dependent manner.

Where does this overabundant CP110 and CEP97 accumulate? Using immunofluorescence microscopy of cycling cells treated with nocodazole, we examined the localization of CP110 and CEP97 to centrioles. In the absence of PCM1, CP110, and CEP97 over-accumulated at both centrioles (*Figure 8F–I*), suggesting that centriolar satellites restrict CP110 and CEP97 accumulation at centrioles.

We conclude that centriolar satellites restrict CP110 and CEP97 levels at centrioles, the removal of which promotes ciliogenesis in specific cell types. Centriolar satellites help promote timely ciliary vesicle formation and remove CP110 and CEP97 from the mother centriole, enabling recruitment of IFT and construction of the transition zone, early steps in ciliogenesis important for the prevention of ciliopathy-associated phenotypes such as hydrocephaly (*Figure 9*).

## Discussion

### PCM1 performs select ciliogenic functions in vivo

Cilia are essential for key events in mammalian development; mice lacking cilia die during embryogenesis with developmental defects including randomized left–right axes and polydactyly (*Ferrante et al., 2006*; *Huangfu et al., 2003*). Many E18.5 *Pcm1⁻/⁻* tissues possessed cilia and *Pcm1⁻/⁻* mice survived at Mendelian ratios to birth and displayed no evidence of situs abnormalities or polydactyly, revealing that centriolar satellites are not required for mammalian ciliogenesis in many cell types.

Despite PCM1 being dispensable for ciliogenesis in many tissues, most *Pcm1⁻/⁻* mice died perinatally with hydrocephaly, delayed formation and disrupted function of ependymal cilia, oligospermia, and abnormalities in tracheal epithelial cell ciliogenesis. In addition, *Pcm1⁻/⁻* mice exhibited cerebellar hypoplasia and partially penetrant hydronephrosis, both of which can be caused by defective Hedgehog signaling, a signal transduction pathway dependent on cilia (*Huangfu et al., 2003*; *Spassky et al., 2008*; *Wallace, 1999*; *Wechsler-Reya and Scott, 1999*; *Yu et al., 2002*). These phenotypes indicate that PCM1 promotes ciliogenesis in select cell types, many of which possess motile cilia.

Recently, a mouse *Pcm1* gene trap was described (*Monroe et al., 2020*). Aged mice homozygous for this allele exhibited enlarged brain ventricles, progressive neuronal cilia maintenance defects and late-onset behavioral changes, but not perinatal lethality or other early cilia-associated phenotypes. While background differences may influence penetrance and expressivity, it is possible that the absence of reported hydrocephaly and other ciliopathy-related phenotypes indicates that the *Pcm1* gene trap allele is hypomorphic.

Most human ciliopathies affect select tissues (*Reiter and Leroux, 2017*). For many ciliopathies, it remains unclear why tissues are differentially sensitive to ciliary defects. As mammalian PCM1 is particularly required for cilia function in ependymal cells and sperm, differential requirements for centriolar satellite function may be one determinant of tissue specificity in human ciliopathies.

### PCM1 and centriolar satellites promote centriole amplification in ependymal cells

In almost all cells, centriole duplication is tightly restricted to make only two new centrioles per cell cycle (*Nigg and Holland, 2018*). In marked contrast, multiciliated cells produce tens to hundreds of centrioles. This centriole amplification has been proposed to occur by two mechanisms: (1) generation of new centrioles in proximity to the parental centrioles and (2) generation via deuterosomes, electron dense structures unique to multiciliated cells (*Mercey et al., 2019b*; *Nanjundappa et al., 2019*; *Zhao et al., 2013*; *Zhao et al., 2019*). However, centriole amplification and multiciliogenesis are not blocked in the absence of deuterosomes or parental centrioles (*Mercey et al., 2019a*; *Mercey et al., 2019b*; *Zhao et al., 2019*), indicating that a third mechanism of centriole biogenesis exists.

A previous study demonstrated that knockdown of *Pcm1* in cultured mouse ependymal cells did not affect centriole number, but did alter ciliary structure (*Zhao et al., 2021*). We found that, in the absence of PCM1, ependymal cells displayed retarded centriole amplification and multiciliogenesis, as well as hydrocephaly. Our data indicate that PCM1, unlike deuterosomes, is critical for timely centriole amplification in ependymal cells. We propose that PCM1 is key to this previously postulated third mechanism of centriole amplification.

Ependymal cells lacking PCM1 also displayed disorganized beat patterns with disrupted basal body translational polarity. In contrast, tracheal multiciliated cells, which do not undergo clear planar

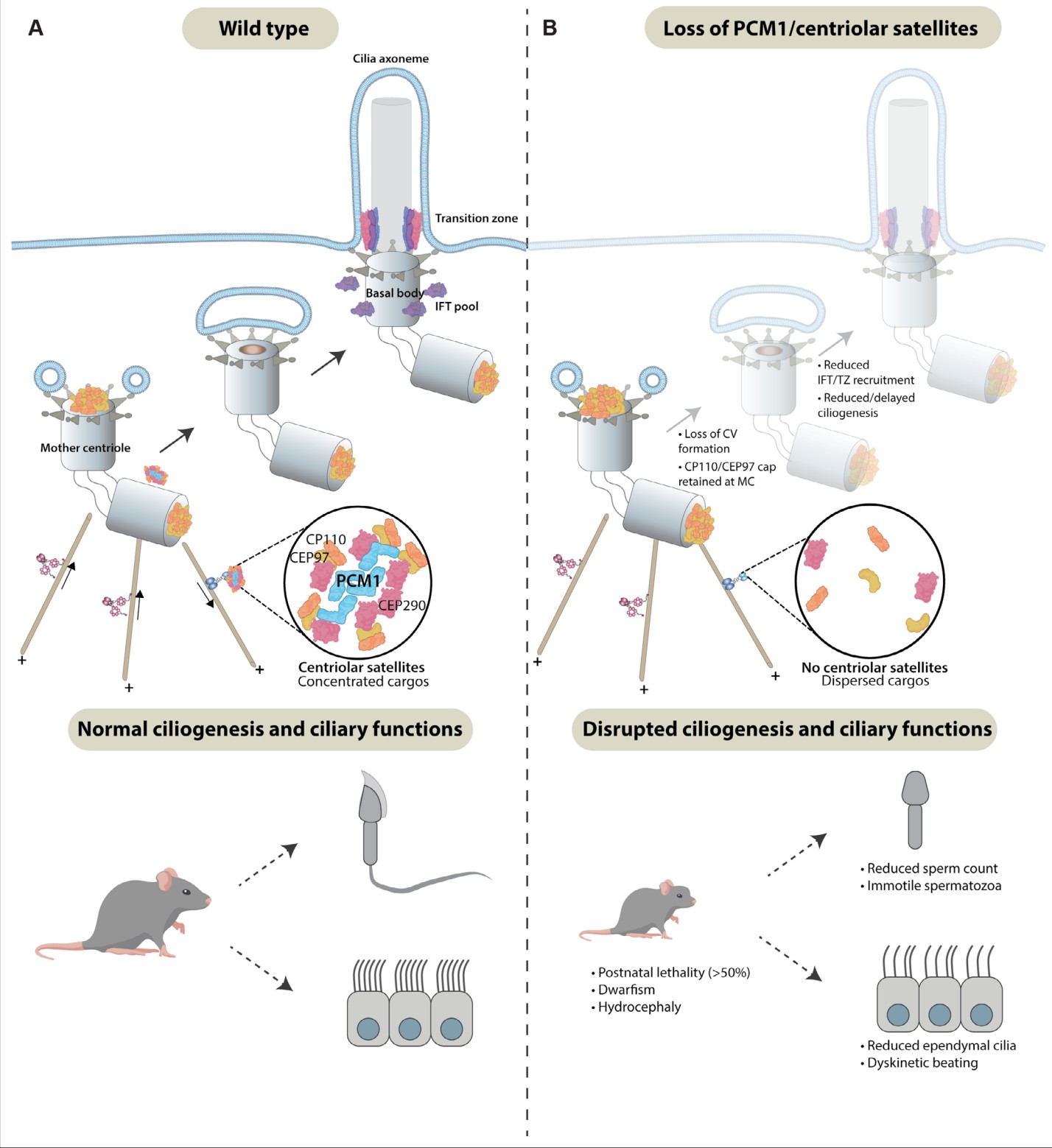

**Figure 9.** Centriolar satellites remodel centrioles to promote ciliogenesis. (**A**) PCM1 (cyan) scaffolds centriolar satellites, dynamic and heterogeneous condensates of centriolar proteins. During ciliogenesis, we propose that centriolar satellites remove, or wick away, CP110 and CEP97 from the mother centriole. Departure of CP110 and CEP97 is important for subsequent steps in ciliogenesis, including centriolar vesicle formation, transition zone formation, and IFT recruitment. (**B**) In the absence of PCM1 and centriolar satellites, CP110 and CEP97 are not efficiently removed during ciliogenesis, disrupting subsequent steps, impeding ciliogenesis in a cell-type-specific way and leading to hydrocephaly and other ciliopathy-associated phenotypes.

polarization of basal body position, displayed normal beating in the absence of PCM1. We speculate that the role of PCM1 in basal body polarization could underlie its unique requirement for beat pattern in ependymal cells. Perhaps the involvement of PCM1 in ependymal cell basal body polarization explains the presence of hydrocephaly in *Pcm1*⁻/⁻ mice with no gross effect on airway mucus clearance.

*Pcm1*⁻/⁻ ependymal cells also generated extremely long (3–7 μm) centriole-like structures containing FOP and Centrin2. These centriole-related structures were present within the cytoplasm, distant from the apical domain where basal bodies nucleate cilia, and are reminiscent of elongated centrioles caused by depletion of CP110 (*Spektor et al., 2007*). One possibility is that the mechanisms by which CP110 and PCM1 restrain elongation are distinct. Alternatively, in multiciliated ependymal cells, the increased CP110 at many basal bodies may deplete the available pool of free CP110, causing a minority of basal bodies to be depleted of CP110 and thereby elongate abnormally.

Consistent with prior observations by *Zhao et al., 2021*, *Pcm1*⁻/⁻ tracheal and ependymal multiciliated cells showed altered fibrogranular material, intracellular networks to which many centriolar proteins localize. Interestingly, PCM1 loss affects fibrogranular material differently in ependymal cells and multiciliated tracheal cells: in *Pcm1*⁻/⁻ mTECs, the fibrogranular material pool of CEP131 is absent and CEP131 accumulates at the basal bodies, whereas in *Pcm1*⁻/⁻ ependymal cells, the fibrogranular material pool of CEP131 persists, but is altered, displaying a more fibrous organization. It is possible that by altering the fibrogranular material, the loss of PCM1 alters the distribution and function of centriolar proteins, resulting in delayed centriole biogenesis and the generation of long centriole-related structures.

## Centriolar satellites promote the timely removal of CP110 and CEP97 to support ciliogenesis

Our work indicates that PCM1 and centriolar satellites help control the composition of centrioles. We found that in diverse cell types, including MEFs, RPE1, ependymal and tracheal cells, PCM1 promotes the removal of CP110 from distal mother centrioles, an early step in ciliogenesis. Similarly, PCM1 restricts levels of CEP131, CEP290, and CEP97 at centrioles. Recent work showed that *Pcm1* knockdown in ependymal cells also increased CEP135 and CEP120 localization to basal bodies (*Zhao et al., 2021*). Thus, centriolar satellites restrict the centriolar accumulation of multiple proteins.

A previous study proposed a role for PCM1 in protecting Talpid3 from degradation by sequestering the E3 ligase, MIB1 away from the centrioles (*Wang et al., 2016*). We found that, in the absence of PCM1, MIB1 no longer localizes to centrioles and Talpid3 levels on *PCM1*⁻/⁻ centrioles were comparable to control centrioles, suggesting that PCM1 is not a critical determinant of centriolar Talpid3 levels. Talpid3 is required for distal appendage assembly and removal of daughter centriole proteins (e.g., Centrobin) from mother centrioles (*Wang et al., 2018*). We found that PCM1 is dispensable for distal appendage assembly and removal of Centrobin from the mother centrioles, further suggesting that PCM1 and centriolar satellites are not required for Talpid3-dependent functions. Thus, centriolar satellites limit the centriolar localization of some, but not all, centriole components.

In the absence of PCM1, total cellular CP110 levels are increased and CP110 and CEP97 levels are elevated at centrioles, indicating a role for centriolar satellites in CP110 degradation. As CP110 and CEP97 transiently localized at satellites, we propose that satellites transport CP110 and CEP97 away from centrioles for degradation. Alternately, satellites could deliver proteins that degrade CP110 and CEP97 to the mother centriole. Such proteins could include UBR5, an E3 ubiquitin ligase that ubiquitinylates CP110, the linear ubiquitin chain assembly complex (LUBAC) that also ubiquitinylates CP110, or PRPF8, which removes ubiquitinylated CP110 from centrioles (*Gonçalves et al., 2021*; *Hossain et al., 2017*; *Shen et al., 2022*). As centriolar satellite composition and distribution can change in response to environmental cues and stressors (*Joachim et al., 2017*; *Prosser et al., 2022*; *Tollenaere et al., 2015*; *Villumsen et al., 2013*), satellites likely help remove centriolar proteins beyond CP110 and CEP97.

The transient localization of CP110 to centriolar satellites is dependent on its interactor, CEP290. As inhibition of ciliogenesis by CP110 is dependent on CEP290 (*Tsang et al., 2008*), we suggest that one function for CEP290 may be to recruit CP110 to satellites for removal from mother centrioles.

In vertebrates, CP110 is required for docking of the mother centriole to preciliary vesicles (*Walentek et al., 2016*; *Yadav et al., 2016*) and is removed from the mother centriole subsequent to

docking, suggesting that CP110 has both positive and inhibitory roles in ciliogenesis (*Lu et al., 2015*). Our finding that PCM1 promotes both CP110 removal and vesicular docking of the mother centriole suggests that centriolar satellites are involved in both intimately connected processes. One possibility is that centriolar satellites promote preciliary vesicle formation via transporting CP110 away from the mother centriole. This possibility is supported by data indicating that WDR8, another centriolar and centriolar satellite component, also contributes to CP110 removal from mother centrioles and ciliary vesicle formation (*Kurtulmus et al., 2016*). However, centriolar satellites may contribute to preciliary vesicle docking through mechanisms independent of CP110 removal. For example, although PCM1 is dispensable for distal appendage formation, the subtle changes in some distal appendage component localization in *PCM1*⁻/⁻ cells could alter distal appendage composition or conformation in ways that compromise preciliary vesicle docking.

Interestingly, despite centriolar satellites promoting removal of CP110 from MEF mother centrioles, they are dispensable for ciliogenesis in MEFs. Therefore, removal of all CP110 from mother centrioles is not a precondition for ciliogenesis in some cell types. Multiple roles for CP110, both promoting and inhibiting ciliogenesis, have previously been described (*Gonçalves et al., 2021*; *Spektor et al., 2007*; *Walentek et al., 2016*; *Yadav et al., 2016*). One possible explanation for the cell-type specificity of PCM1 function is that centriolar satellites remove CP110 from mother centrioles in all cell types, but different thresholds of CP110 reduction are required to initiate ciliogenesis in different cell types. Thus, unlike core centriolar proteins, some of which are trafficked via centriolar satellites, centriolar satellites themselves are not essential for all centriole- and cilium-dependent events in many mammalian cell types.

These cell-type-specific differences may reflect differences in how centrioles must be remodeled to effect duplication or ciliogenesis. Perhaps centriolar satellite-mediated CP110 removal from mother centrioles is especially important for cells, like many epithelial cells, in which basal bodies dock directly to the plasma membrane, rather than to a ciliary vesicle. In the crowded environment at the heart of the centrosome, diffusion may be insufficient for the timely delivery and removal of centriolar proteins. PCM1 and centriolar satellites promote centriole amplification and ciliogenesis by coupling assembly and/or degradation of centriolar components in the satellites to their active transport to and from centrioles on microtubules.

## Materials and methods

### Generation of mouse models

Animals were maintained in SPF environment and studies carried out in accordance with the guidance issued by the Medical Research Council in 'Responsibility in the Use of Animals in Medical Research' (July 1993) and licensed by the Home Office under the Animals (Scientific Procedures) Act 1986 under project license number P18921CDE in facilities at the University of Edinburgh (PEL 60/6025). *Pcm1* null mice (*Pcm1*^Δ5-14/Δ5-14: *Pcm1*^em1Pmi MGI:6865681 and *Pcm1*^Δ796-800/Δ796-800: *Pcm1*^em2Pmi MGI:6865682) were generated using CRISPR/Cas9 as described in *Figure 1—figure supplement 1*, using guides detailed in *Supplementary file 1*. Genotyping was performed using primers detailed in *Supplementary file 2* followed by Sanger sequencing (for *Pcm1*^Δ5-14/Δ5-14) or digestion with DdeI (for *Pcm1*^Δ796-800/Δ796-800), or alternately genotyping was performed by Transnetyx. *Pcm1*^SNAP animals were generated with CRISPR Cas9 targeting first coding exon 2 (*Supplementary file 1*) and a SNAP tag was inserted after the ATG, followed by a GSGG linker, using a repair template with 700 nt homology arms, detailed in *Supplementary file 1*, resulting in a gene encoding N-terminally SNAP tagged PCM1 in the endogenous locus. Genotyping was performed using primers detailed in *Supplementary file 2* or alternately by Transnetyx.

### Mouse gait analysis

Gait analysis was performed on a Catwalk XT according to manufacturer's instructions. Briefly, mice were habituated to the Catwalk for 5 min, and then the glass was cleaned prior to acquisition. Each mouse ($n > 4$ per experimental group) was then allowed to perform at least 3 runs across the Catwalk, which records paw position and analyses gait patterns using the Catwalk XT 10.6 Acquisition and Analysis Software.

## Retinal imaging

Electroretinograms and fundal imaging was performed as described in *Findlay et al., 2018*. PCM1-SNAP retinal labeling was carried out under inhaled anesthesia. 1.5 µl of 0.6 µM SNAP-Cell 647-SiR (New England Biolabs) was injected into the mouse vitreous under direct visualization using a Zeiss operating microscope. After 2 hr, mice were sacrificed by cervical dislocation and eyes enucleated. Keratectomy, sclerectomy and lensectomy were performed and whole retinas isolated. Flat mount petaloid retinal explants were made and mounted, photoreceptor side up, on Menzel_Glaser Superfrost Plus Gold slides (Thermo Fisher Scientific; K5800AMNZ72). Nuclei were stained with DAPI and mounted in Prolong Gold under coverslip. Slices were imaged on an Andor Dragonfly spinning disc confocal.

## Cell lines and cell culture

MEFs were maintained as previously published (*Hall et al., 2013*). SNAP labeling was performed as previously described (*Quidwai et al., 2021*). Ependymal cells were isolated and cultured as published in *Delgehyr et al., 2015*. mTECs were isolated and cultured as described in *Eenjes et al., 2018*; *You et al., 2002*. RPE1-hTERT (female, human epithelial cells immortalized with hTERT, Cat. No. CRL-4000) from ATCC were grown in Dulbecco's Modified Eagle Medium (DMEM, Life Technologies) or DMEM/F12 (Thermo Fisher Scientific, 10565042) supplemented with 10% fetal bovine serum at 37°C with 5% $CO_2$. For live imaging, the membrane was cut out and placed cilia down on a glass dish (Nest, 801002) in a drop of media. *PCM1$^{-/-}$* RPE1 cells were generated as described previously (*Kumar et al., 2021*) (all figures except for *Figure 8—figure supplement 1*, in which case they were generated as in *Gheiratmand et al., 2019*). hTERT-RPE1: Source ATCC, confirmed mycoplasma negative and verified by STR profiling. Two *PCM1$^{-/-}$* RPE1 cell lines were generated using single guide RNAs (*Supplementary file 1*). Loss of PCM1 was confirmed by genotyping, immunoblotting, and immunofluorescence. Monoclonal *PCM1$^{-/-}$* RPE1 cell lines stably expressing eGFP or eYFP-PCM1 (plasmid a gift from Bryan Dynlacht; *Wang et al., 2016*) were generated using lentiviruses and manually selected based on fluorescence. To synchronize cells in G1/S aphidicolin (Sigma) was added to the culture medium at 2 µg/ml for 16 hr. To arrest cells in mitosis, taxol (paclitaxel; Millipore-Sigma) was added to the culture medium at 5 µM for 16 hr prior to rounded up cells being collected by mitotic shake-off. For arrest in G0, cells were washed 2× with phosphate-buffered saline (PBS; Gibco) and 1× with DMEM (without serum) before being cultured in serum-free DMEM for 16 hr. To disrupt cytoplasmic microtubules, cells were treated with 20 µM nocodozole (Sigma, SML1665) for 1–2 hr prior to fixation.

## RNA-mediated interference

hTERT RPE-1 cells were transfected with 20 nM (final concentration) of the respective siRNA for 48 hr using Lipofectamine RNAiMAX (Invitrogen) according to the manufacturer's instructions. Effective knockdown was confirmed by immunofluorescence microscopy. Details of individual siRNAs are provided in the *Supplementary file 1*.

## Proteomics

mTECs were lysed in 0.1% sodium dodecyl sulfate (SDS) in PBS plus 1× HALT protease inhibitor (Thermo Fisher Scientific, 78443), then processed by a multi-protease FASP protocol as described (*Wiśniewski and Mann, 2012*). In brief, SDS was removed and proteins were first digested with Lys-C (Wako) and subsequently with Trypsin (Promega) with an enzyme to protein ratio (1:50). 10 µg of Lys-C and Trypsin digests were loaded separately and desalted on C18 Stage tip and eluates were analyzed by high-performance liquid chromatography coupled to a Q-Exactive mass spectrometer as described previously (*Farrell et al., 2014*). Peptides and proteins were identified and quantified with the MaxQuant software package, and label-free quantification was performed by MaxLFQ (*Cox et al., 2014*). The search included variable modifications for oxidation of methionine, protein N-terminal acetylation, and carbamidomethylation as fixed modification. Peptides with at least seven amino acids were considered for identification. The false discovery rate (FDR), determined by searching a reverse database, was set at 0.01 for both peptides and proteins. All bioinformatic analyses were performed with the Perseus software (*Tyanova et al., 2016*). Intensity values were log-normalized, 0-values were imputed by a normal distribution 1.8 π down of the mean and with a width of 0.2 π.

Proteomic expression data were analyzed in R (3.6.0) with the Bioconductor package DEP (1.6.1) (*Zhang et al., 2018*). To aid in the imputation of missing values only those proteins that are identified in all replicates of at least one condition were retained for analysis. The filtered proteomic data were normalized by variance stabilizing transformation. Following normalization, data missing at random, such as proteins quantified in some replicates but not in others, were imputed using the *k*-nearest neighbour approach. For differential expression analysis between the wild-type and mutant groups, protein-wise linear models combined with empirical Bayes statistics were run using the Bioconductor package limma (3.40.6) (*Ritchie et al., 2015*). Significantly differentially expressed proteins were defined by an FDR cutoff of 0.05. Total proteomic data are available via ProteomeXchange with identifier PXD031920 and are summarized in *Supplementary file 5*.

## Immunoblotting

Testes were lysed in RIPA buffer (Pierce) plus HALT protease inhibitor (Thermo Fisher Scientific), homogenized with an electronic pestle for 1 min, incubated at 4°C with agitation for 30 min, sonicated for 3 × 30 s, and then clarified at 14,000 × *g* at 4°C for 20 min. RPE lysates were collected in 2× SDS–polyacrylamide gel electrophoresis (PAGE) buffer and treated with benzonase nuclease (Millipore-Sigma) for 5 min. Samples were loaded into NuPAGE precast gels, transferred onto polyvinylidene fluoride (PVDF) membrane (Amersham Hybond P, Cytiva), and then rinsed in water then TBST, and then blocked in 5% milk in TBS plus 0.1% Tween. Membranes were then incubated overnight at 4°C in primary antibodies (*Supplementary file 3*) diluted in 5% milk TBST. Membranes were then washed 3 × 10 min TBST, incubated in Horse Radish Peroxidase (HRP)-conjugated secondary antibodies detailed in *Supplementary file 4* for 1 hr at room temperature and developed using Pierce SuperSignal Pico Plus (Pierce) or ECL (GE Healthcare) reagent and imaged on ImageQuant.

## Co-immunoprecipitation

Co-immunoprecipitation assays and western blots were performed as described previously (*Kumar et al., 2021*) using GFP trap magnetic agarose beads (Chromotek, gtma-10).

## Ventricle and tracheal wholemount

Ventricles were dissected according to *Mirzadeh et al., 2010a*, pre-extracted with 0.1% Triton X in PBS for 1 min, then fixed in 4% paraformaldehyde (PFA) or ice cold methanol for at least 24 h at 4°C, followed by permeabilization in PBST (0.5% Triton X-100) for 20-min room temperature. Tracheas were dissected and cut longitudinally into two, pre-extracted in for 30 s on ice in PEM (0.1 M PIPES (1,4-Piperazinediethanesulfonic acid disodium salt) pH 6.8, 2 mM EGTA (ethylene glycol tetraacetic acid), 1 mM MgSO$_4$) prior to fixing in ice cold methanol on ice for at least 24 hr. Ventricles and tracheas were blocked in 10% donkey serum in TBST (0.1% Triton X) or 4% bovine serum albumin (BSA) in PBST (0.25% Triton X-100) for 1 hr at room temperature, then placed cilia layer down in primary antibodies (*Supplementary file 3*) in 4% BSA PBST (0.25% Tween-20) or 1% donkey serum in TBST (0.1% Triton X) for at least 12 hr. Ventricles and tracheas were washed in PBS 3 × 10 min and secondaries (*Supplementary file 4*) in 4% BSA in PBST (0.25% Triton X-100) or 1% donkey serum in TBST (0.1% Triton X) were added at 4°C for at least 12 hr. Ventricles and tracheas were washed in PBS 3 × 10 min, and ventricles were mounted on glass bottom dishes (Nest, 801002) in Vectashield (VectorLabs), immobilized with a cell strainer (Greiner Bio-One, 542040). Tracheas were mounted on slides with Prolong Gold.

## Histology

Kidneys and brains were fixed in 4% PFA/PBS, testes were fixed in Bouin's fixative, and eyes and E18.5 embryos were fixed in Davidson's fixative according to standard protocols. Tissues were serially dehydrated and embedded in paraffin. Microtome sections of 8 µm thickness were examined histologically via haematoxylin and eosin (H&E) or periodic acid-Schiff (PAS) staining.

For immunofluorescent analysis, paraffin sections were dewaxed and re-hydrated via ethanol series, followed by antigen retrieval by boiling the sections for 15 min in the microwave in citrate buffer. Sections were blocked in 10% donkey serum/0.1% Triton X-100 in PBS and primary antibodies were diluted in 1% donkey serum/PBS (*Supplementary file 3*). Slides were washed and incubated in Alexafluor conjugated secondary antibodies (*Supplementary file 4*), washed and mounted in ProLong Gold (Thermo Fisher Scientific).

## Immunofluorescence

MEFs, mTECs, and cultured ependymal cells were processed for immunofluorescence as published (*Hall et al., 2013*). Briefly, cells were washed twice with warm PBS, then fixed in either 4% PFA in 1× PHEM (PIPES pH 6.9, HEPES (-2-hydroxyethylpiperazine-N-2-ethane sulfonic acid), EGTA, MgCl2)/ PBS 10 min at 37°C, or pre-extracted for 30 s on ice in PEM (0.1 M PIPES pH 6.8, 2 mM EGTA, 1 mM MgSO$_4$) prior to fixing in ice cold methanol on ice for 10 min according to *Supplementary file 3*, then washed twice with PBS. Cells were permeabilized and blocked with 10% donkey serum in 0.1% Triton X-100/TBS for 60 min at room temperature, or overnight at 4°C. Primary antibodies (*Supplementary file 3*) were added to samples and incubated for 4°C overnight, in dilutant made of 1% donkey serum in 0.1% Triton X-100/TBS. Samples were washed in 0.1% Triton X-100/TBS 4–6 times, 10 min each. Secondary antibodies (*Supplementary file 4*) diluted in 1% donkey serum and 0.1% Triton X-100/TBS were added for 60 min at room temperature, in some cases co-stained with AlexaFluor 647 Phalloidin (Thermo Fisher Scientific), added with the secondaries at 1/500 for 1 hr at room temperature. Samples were washed with 0.1% Triton X-100/TBS 4–6 times 10 min, stained with DAPI (1:1000) in 0.1% Triton X-100/TBS for 5 min at room temperature, and mounted using ProLong Gold antifade (Thermo Fisher Scientific), according to the manufacturer's instructions.

RPE1 cells were fixed with 100% cold methanol for 3 min and incubated in blocking buffer (2.5% BSA, 0.1% Triton X-100 in PBS) for 1 hr at room temperature (except in *Figure 8—figure supplement 1* where they were fixed in ice cold methanol for 10 min and incubated in 2% BSA in PBS for 10 min at room temperature). Coverslips were then incubated in primary antibodies (*Supplementary file 3*) in blocking buffer overnight at 4°C or room temperature for 50 min, washed three times with PBS and incubated with secondary antibodies (*Supplementary file 4*) in blocking buffer for 1 hr at room temperature along with Hoechst 33352 or DAPI (0.1 µg/ml). Coverslips were washed three times with PBS and mounted with Prolong Diamond (Thermo Fisher Scientific P36961) or ProLong Gold Antifade (Molecular Probes). For TTBK2 staining, cells were fixed with 4% PFA/PBS for 10 min in general tubulin buffer (80 mM PIPES, pH 7, 1 mM MgCl$_2$, and 1 mM EGTA), permeabilized with 0.1% TX-100 and stained as described above (*Loukil et al., 2017*).

## Sperm preparation

Cauda and caput epididymides were dissected into M2 media (Thermo Fisher Scientific). For live imaging, sperm were imaged in M2 media or 1% methyl cellulose (Sigma), in capillary tubes (Vitro-tubes Mountain Leaks) sealed with Cristaseal (Hawskley). Sperm counts were performed on sperm from the cauda epididymides, diluted in H$_2$O using a haemocytometer, only counting intact sperm (with both head and tail).

## Transmission electron microscopy

Samples were dissected into PBS. Samples were fixed in 2% PFA/2.5% glutaraldehyde/0.1 M sodium cacodylate buffer pH 7.4 (Electron Microscopy Sciences). Lateral ventricle walls were fixed for 18 hr at 4°C then subdissected into anterior, mid, and posterior sections. Tissue was rinsed in 0.1 M sodium cacodylate buffer, post-fixed in 1% OsO$_4$ (Agar Scientific) for 1 hr and dehydrated in sequential steps of acetone prior to impregnation in increasing concentrations of resin (TAAB Lab Equipment) in acetone followed by 100%, placed in moulds and polymerized at 60°C for 24 hr.

Ultrathin sections of 70 nm were subsequently cut using a diamond knife on a Leica EM UC7 ultra-microtome. Sections were stretched with chloroform to eliminate compression and mounted on Pioloform filmed copper grids prior to staining with 1% aqueous uranyl acetate and lead citrate (Leica). They were viewed on a Philips CM100 Compustage Transmission Electron Microscope with images collected using an AMT CCD camera (Deben).

RPE1 cells processed for TEM analysis were cultured on Permanox slides (Nunc 177445), serum starved for 1 hr and processed as described previously (*Kumar et al., 2021*).

## Imaging

Brightfield images in *Figure 2* and *Figure 2—figure supplement 1* were imaged on a Hamumatsu Nanozoomer XR with ×20 and ×40 objectives. Macroscope images in *Figure 1* and *Figure 2* were imaged on a Nikon AZ100 Macroscope. *Figure 1—figure supplement 3* was imaged on Leica Stellaris DMI8 equiped with 4 (HyD X/HyD S) GaSP detectors with ×40 or ×60 oil objectives. Fluorescent

images in *Figure 2*, *Figure 3A*, *Figure 3—figure supplement 1A, B*, *Figure 3—figure supplement 2A*, *Figure 3—figure supplement 3*, and *Figure 7—figure supplement 1* were taken on a Nikon A1+Confocal with Oil 60 or ×100 objectives with 405, Argon 561 and 640 lasers and GaSP detectors. Fluorescent images in *Figure 1*, *Figure 2—figure supplement 1D*, *Figure 4D, E, K*, *Figure 4—figure supplement 1*, and *Figure 6—figure supplement 1* were taken with Andor Dragonfly and Mosaic Spinning Disc confocal. Images in *Figure 3B, O*, *Figure 3—figure supplement 1C, H, I*, *Figure 3—figure supplement 2C–E*, and *Figure 6E, F* were taken with Nikon SORA with 405 nm 120 mW, 488 nm 200 mW, and 561 nm 150 mW lasers, ×100 1.35 NA Si Apochromat objective and a Photometrics Prime 95B 11 mm pixel camera. High-speed video microscopy was performed on a Nikon Ti microscope with a ×60 Nikon Plan Apo VC ×60/1.20 water immersion objective, and Prime BSI, A19B204007 camera, imaged at 250 fps. 3D-SIM imaging in *Figure 4B, C, H*, *Figure 5*, *Figure 5—figure supplement 1*, *Figure 6A, B*, *Figure 7*, and *Figure 8* was performed using the GE Healthcare DeltaVision OMX-SR microscope equipped with the ×60/1.42 NA oil-immersion objective and three cMOS cameras. Immersion oil with refractive index of 1.518 was used for most experiments, and z stacks of 5–6 μm were collected every 0.125 μm. Images were reconstructed using GE Healthcare SoftWorx 6.5.2 using default parameters. Images for quantifications were collected at the widefield setting using the same microscope. *Figure 8—figure supplement 1* was imaged using a DeltaVision Elite high-resolution imaging system equipped with a sCMOS 2048x2048 pixel camera (GE Healthcare). Z-stacks (0.2 μm step) were collected using a ×60 1.42 NA plan apochromat oil-immersion objective (Olympus) and deconvolved using softWoRx (v6.0, GE Healthcare).

## Image analysis

Image analysis was performed in NIS Elements, FIJI (*Schindelin et al., 2012*), QuPath (*Bankhead et al., 2017*), CellProfiler (*Stirling et al., 2021*), or Imaris. All analysis tools have been made available on GitHub (https://github.com/IGC-Advanced-Imaging-Resource/Hall2022_Paper; *Murphy, 2022*). Cerebellum and ventricle area was measured from PAS stained sagittal brain sections in QuPath. The number of cilia in E18.5 ribs was calculated using Batch Pipeline in Imaris, segmenting DAPI and cilia as surfaces. The number of ependymal cells with multiple basal bodies was calculated by segmenting FOP staining and cells in 2D using a CellProfiler pipeline. Briefly, an IdentifyPrimaryObjects module was used to detect the nuclei, followed by an IdentifySecondaryObjects module using the tubulin stain to detect the cell boundaries. Another Identify Primary objects module was used to detect the basal bodies and a RelateObjects module was used to assign parent–child relationships between the cells and basal bodies. The percentage of ciliated ependymal cells, and the number of ependymal cells with rosette-like FOP staining, and elongated FOP-positive structures were counted by eye using NIS Elements Counts Tool. Analysis of cultured ependymal cells (beat frequency, number of cilia, coordinated beat pattern) and beat frequency determination in mTECs and trachea was assessed in FIJI by eye while blinded to genotype. The number of centrioles and cilia in cultured ependymal cells was manually calculated using Imaris. CEP131 and MIB1 intensity at satellites was calculated in FIJI using a macro which segmented basal bodies with Gamma Tubulin, then drew concentric rings, each 0.5 μm wider than the previous and calculated the intensity of MIB1 and CEP131 within these rings. CP110 intensity in MEFs was calculated by manually defining mother and daughter centrioles in FIJI, CP110 intensity in ependyma and tracheas was calculated by segmenting FOP in 3D in Imaris and calculating CP110 intensity within this volume. Image quantification in RPE1 cells were performed using CellProfiler as described previously (*Kumar et al., 2021*). Images were prepared for publication using FIJI, Imaris, Adobe Photoshop, Illustrator, and InDesign.

## Data analysis

Data analysis was carried out in Microsoft Excel, GraphPad Prism 6/9, and Matlab. Statistical tests are described in the figure legends.

## Acknowledgements

We thank the IGC Advanced Imaging Resource and the IGC Mass Spectrometry facility, as well as the Newcastle University Electron Microscopy Research services and, in particular, Tracey Davey. Work by the group of JFR is supported by NIH R01GM095941, R01AR054396, and R01HD089918. DK is supported by NIH K99 grant (5K99GM140175) and was funded by the Jane Coffin Childs memorial

foundation and a Program for Biomedical Research award by the Sandler foundation. Work by the group of PM is supported by MRC intramural funding (MC_UU_12018/26) and by the European Commission (H2020 Grant No. 866355). Work by the group of LP is supported by CIHR Foundation (FDN#167279) and the Krembil Foundation. LP is a Tier 1 Canada Research Chair in Centrosome Biogenesis and Function and SLP was funded by a European Union Horizon 2020 Marie Skłodowska-Curie Global Fellowship (No. 702601).

## Additional information

### Competing interests

Jeremy F Reiter: Reviewing editor, eLife. The other authors declare that no competing interests exist.

### Funding

| Funder | Grant reference number | Author |
|---|---|---|
| Medical Research Council | MR_UU_1201018/26 | Emma A Hall<br>Dhivya Kumar<br>Patricia L Yeyati<br>Lorraine Rose<br>Lisa McKie<br>Daniel O Dodd<br>Peter A Tennant<br>Roly Megaw<br>Laura C Murphy<br>Graeme Grimes<br>Lucy Williams<br>Tooba Quidwai<br>Pleasantine Mill<br>Marisa F Ferreira |
| European Commission | 866355 | Emma A Hall<br>Daniel O Dodd<br>Pleasantine Mill |
| Canadian Institutes of Health Research | 167279 | Suzanna L Prosser<br>Laurence Pelletier |
| National Institutes of Health | R01GM095941 | Vicente Herranz-Pérez<br>Jose Manuel García-Verdugo<br>Jeremy F Reiter<br>Dhivya Kumar |
| Jane Coffin Childs Memorial Fund for Medical Research | | Dhivya Kumar |
| Sandler Foundation | | Dhivya Kumar |
| Krembil Foundation | | Suzanna L Prosser<br>Laurence Pelletier |
| European Commission | 702601 | Suzanna L Prosser |
| National Institutes of Health | R01AR054396 | Dhivya Kumar<br>Vicente Herranz-Pérez<br>Jose Manuel García-Verdugo<br>Jeremy F Reiter |
| National Institutes of Health | RO1HD089918 | Vicente Herranz-Pérez<br>Jose Manuel García-Verdugo<br>Jeremy F Reiter<br>Dhivya Kumar |
| National Institutes of Health | 5K99GM140175 | Dhivya Kumar |

| Funder | Grant reference number | Author |
|---|---|---|

The funders had no role in study design, data collection, and interpretation, or the decision to submit the work for publication.

## Author contributions

Emma A Hall, Conceptualization, Formal analysis, Supervision, Investigation, Methodology, Writing – original draft; Dhivya Kumar, Conceptualization, Formal analysis, Investigation, Methodology, Writing – original draft; Suzanna L Prosser, Formal analysis, Investigation, Writing – review and editing; Patricia L Yeyati, Investigation, Methodology, Writing – review and editing; Vicente Herranz-Pérez, Jose Manuel García-Verdugo, Roly Megaw, Investigation, Methodology; Lorraine Rose, Lisa McKie, Resources; Daniel O Dodd, Peter A Tennant, Investigation, Writing – review and editing; Laura C Murphy, Software, Formal analysis, Methodology; Marisa F Ferreira, Formal analysis, Investigation; Graeme Grimes, Formal analysis, Visualization; Lucy Williams, Investigation; Tooba Quidwai, Methodology, Writing – review and editing; Laurence Pelletier, Supervision, Writing – review and editing; Jeremy F Reiter, Conceptualization, Supervision, Writing – original draft, Project administration; Pleasantine Mill, Conceptualization, Supervision, Funding acquisition, Writing – original draft, Project administration

## Author ORCIDs

Emma A Hall http://orcid.org/0000-0002-6944-6719
Dhivya Kumar http://orcid.org/0000-0002-3737-014X
Vicente Herranz-Pérez http://orcid.org/0000-0002-1969-1214
Roly Megaw http://orcid.org/0000-0001-5605-4540
Marisa F Ferreira http://orcid.org/0000-0002-8123-4612
Tooba Quidwai http://orcid.org/0000-0001-5248-9010
Laurence Pelletier http://orcid.org/0000-0003-1171-4618
Jeremy F Reiter http://orcid.org/0000-0002-6512-320X
Pleasantine Mill http://orcid.org/0000-0001-5218-134X

## Ethics

Animals were maintained in SPF environment and studies carried out in accordance with the guidance issued by the Medical Research Council in 'Responsibility in the Use of Animals in Medical Research' (July 1993) and licensed by the Home Office under the Animals (Scientific Procedures) Act 1986 under project license number P18921CDE in facilities at the University of Edinburgh (PEL 60/6025).

## Decision letter and Author response

Decision letter https://doi.org/10.7554/eLife.79299.sa1
Author response https://doi.org/10.7554/eLife.79299.sa2

# Additional files

## Supplementary files

- Supplementary file 1. gRNAs, repair template and siRNA sequences.
- Supplementary file 2. Genotyping primer sequences.
- Supplementary file 3. Primary antibodies.
- Supplementary file 4. Secondary antibodies.
- Supplementary file 5. Differentially expressed proteins between wild-type and *Pcm1*$^{-/-}$ mouse tracheal epithelial cells (mTECs) on at least one timepoint (ALI1, ALI7, and/or ALI21). Expression is given as label-free quantitative (LFQ) normalized by variance stabilizing transformation as described in Materials and Methods, significantly differentially expressed proteins were defined by a false discovery rate (FDR) cutoff of 0.05.
- Transparent reporting form

## Data availability

Proteomics data files are uploaded to ProteomeXchange (Identifier: PXD031920), with the accession number is available with the paper. All analysis tools have been made available on GitHub (copy

archived at swh:1:rev:7b4d68b8ba0c7bf5cc06de6b6589656c3785e6e0), as described in Materials and methods.

The following dataset was generated:

| Author(s) | Year | Dataset title | Dataset URL | Database and Identifier |
|---|---|---|---|---|
| Hall EA, Mill P | 2023 | Centriolar satellites expedite mother centriole remodeling to promote ciliogenesis | https://www.ebi.ac.uk/pride/archive/projects/PXD031920 | PRIDE, PXD031920 |

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

# Appendix 1

## Appendix 1—key resources table

| Reagent type (species) or resource | Designation | Source or reference | Identifiers | Additional information |
|---|---|---|---|---|
| Genetic reagent (*M. musculus*) | *Pcm1$^{Δ5-14}$Pcm1$^{em1Pmi}$* MGI:6865681 | This paper | Allele symbol: *Pcm1$^{em1Pmi}$* Allele synonym: *Pcm1$^{Δ5-14}$*; Accession ID: MGI:6865681 | |
| Genetic reagent (*M. musculus*) | *Pcm1$^{Δ796-800}$Pcm1$^{em2Pmi}$* MGI:6865682 | This paper | Allele symbol: *Pcm1$^{em2Pmi}$* Allele synonym: *Pcm1$^{Δ796-800}$*; Accession ID: MGI:6865681 | |
| Genetic reagent (*M. musculus*) | *Pcm1$^{SNAP}$Pcm1$^{em3Pmi}$* MGI:6865681 | This paper | Allele symbol: *Pcm1$^{em3Pmi}$* Allele synonym: *Pcm1$^{SNAP}$*; Accession ID: MGI:6865681 | |
| Biological sample (*M. musculus*) | Mouse embryonic fibroblasts (MEFs) | This paper | N/A | |
| Biological sample (*M. musculus*) | Mouse tracheal epithelial cells (mTECs) | This paper | N/A | See *Vladar and Brody, 2013* for protocol. |
| Biological sample (*M. musculus*) | *PCM1$^{−/−}$* RPE 1 | *Kumar et al., 2021* | | All Figures except *Figure 8—figure supplement 1* |
| Biological sample (*M. musculus*) | *PCM1$^{−/−}$* RPE 1 | *Gheiratmand et al., 2019* | | In *Figure 8—figure supplement 1* |
| Sequence-based reagent | Pcm1 Exon 2 | Dharmacon | 5'-ATTAAAGGCAACATGGCCAC-3' | RISPR guide for generation of Pcm1$^{5-14}$ and Pcm1SNAP mouse |
| Sequence-based reagent | *Pcm1* Exon 6 | Dharmacon | 5'-TCAGGCCAGAGATCCTCAGC-3' | CRISPR guide for generation of *Pcm1 796–800* mouse |

*Appendix 1 Continued on next page*

*Appendix 1 Continued*

| Reagent type (species) or resource | Designation | Source or reference | Identifiers | Additional information |
|---|---|---|---|---|
| Sequence-based reagent | SNAP repair | IDT | 5'- aaaataattctgaagccaaaaaccgct gcaaggaggatttatgagtttggcagacttca gggagattgacacaacactatgagagacagta agcactcattgaaatgtgtttagtgcatttgt tctgttttatttggaacaaactttattttaaatagc ttactataagctcaggctggtctagaacacct gattctcatacttacctccta gtactgcgattataagcatgtgctaccatctc cattatataatgtgtatatcatgtagatcaatttat ctgtgatacgtgtttgatagtgtattctttatatt tttggttgtgagcctagcctttaacagctgag ccatctctccagctcgatagtgtattctttaa gataagtgtttgaaagattcctttatattaat aagtttgatagaatgctttaaaatctgaa gatggttcagcatatgaaagtgcttgccatac aaacctgatgacctcagatcacacagtggcag gagagaactgactccagatagttgctctgacc tctgcacacatgctatggtacatacatgtctg cacttacatacaaaaacatgcatatacacaat ataattattagtacattttataataaaataaagttt gtctttctgtgttaaaaattaatttttacttatttt gcagAGAATTAATTAAAGGCAACA TGGACAAAGACTGCGAAATG AAGCGCACCACCCTGGATAG CCCTCTGGGCAAGCTGGAAC TGTCTGGGTGCGAACAGGGC CTGCACCGTATCATCTTCCTGGGC AAAGGAACATCTG CCGCCGACGCCGTGGAAGTG CCTGCCCCAGCCGCCGTGCT GGGCGGACCAGAGCCACTGA TGCAGGCCACCGCCTGGCTC AACGCCTACTTTCACCAGCCTGAG GCCATCGAGGAGTTCCCTGT GCCAGCCCTGCACCACCCAG TGTTCCAGCAGGAGAGCTTT ACCCGCCAGGTGCTGTGGAA ACTGCTGAAAGTGGTGAAGT TCGGAGAGGTCATCAGCTAC AGCCACCTGGCCGCCCTGGC CGGCAATCCCGCCGCCACCG CCGCCGTGAAAACCGCCCTG AGCGGAAATCCCGTGCCCAT TCTGATCCCCTGCCACCGGG TGGTGCAGGGCGACCTGGAC GTGGGGGGCTACGAGGGCGG GCTCGCCGTGAAAGAGTGGC TGCTGGCCCACGAGGGCCAC AGACTGGGCAAGCCTGGGCT GGGTGGCGGAAGCGGAGCCA CAGGAGGAGGTCCTTTTGAA GAAGTCATGCATGATCAGGACTTA CCAAACTGGAGCAATGACAG TGTGGATGACCGACTCAACAATAT GGTATGATGTTttactctgggtggtata ttgttgaccactaatgttcagtgaggctctcc catcgattgtatttactgaaactctgtaaaaa ctgtaggcagatagactaagggactcttggtt gaagacactttagctgtagttaatagaaagca tgaattagcttaaacaaaaaatgatttattaa aaggaggtgaaagtgctttatggaagccatgt taaagagtatagctcagttttaggaaaggaaa aagaaacagcagagttgttcgaaattgctttt cacctctgtgcctgtgcttctaagaccttttcccta accgagctttcccttctagatctgccttctttctct ctctgctttgtgtcatatattg agatggccttttttaaagatttgcagccatgga ggaacttatataatgactaatttaacattatgatta tctagctaa atttgtttagatctcctttttttcacttatcaggatc atgaaagggatgaattaaataa tataaaaggttcacaggactacccatacatgg aacagttcctcgaggggcaaaatttcctagaa gtgatgacagtactaagcagtttttattatag- 3' | Repair template for generation of Pcm1SNAP mouse |
| Sequence-based reagent | *PCM1* Exon 3 | Synthego | 5'-GAAAAGAAUAAGAAAAAGUU-3' | CRISPR guide for generation of *PCM1*⁻/⁻ RPE cells |
| Sequence-based reagent | *PCM1* Exon 3 | Synthego | 5'-CGACUCCGGAGAAAUAUCA-3' | CRISPR guide for generation of *PCM1*⁻/⁻ RPE cells |
| Sequence-based reagent | Luciferase GL2 Duplex siRNA | Dharmacon | 5'-CGUACGCGGAAUACUUCGA-3' | Control siRNA |

*Appendix 1 Continued on next page*

*Appendix 1 Continued*

| Reagent type (species) or resource | Designation | Source or reference | Identifiers | Additional information |
|---|---|---|---|---|
| Sequence-based reagent | *CEP290* ID: s37024 Silencer Select siRNA | Ambion/Thermo Fisher | 5'-GAUACUCGGUUUUUACGUA-3' | *CEP290* siRNA |
| Sequence-based reagent | *CEP290* ID: s37025 Silencer Select siRNA | Ambion/Thermo Fisher | 5'-CACUUACGGACUUCGUUAA-3' | *CEP290* siRNA |
| Sequence-based reagent | *Pcm1* 2F | Sigma | 5' CTCTGACCTCTGCACACATG 3' | Genotyping *Pcm1*$^{\Delta 5-14}$ mouse. PCR followed by Sanger sequencing. Product size: 332 bp |
| Sequence-based reagent | *Pcm1* 2R | Sigma | 5' ACAATCGATGGGAGAGCCTC 3' | Genotyping *Pcm1*$^{\Delta 5-14}$ mouse. PCR followed by Sanger sequencing. Product size: 332 bp |
| Sequence-based reagent | *Pcm1* 6F | Sigma | 5' AGTATCGCTGTACTTTGCCA 3' | Genotyping *Pcm1*$^{\Delta 796-800}$ mouse. PCR followed by Dde1 digestion. Product size: 266 bp |
| Sequence-based reagent | *Pcm1* 6R | Sigma | 5' CAGAGTCATCCATCACAGCTA T 3' | Genotyping *Pcm1*$^{\Delta 796-800}$ mouse. PCR followed by Dde1 digestion. Product size: 266 bp |
| Sequence-based reagent | *Pcm1* 2F | Sigma | 5' CTCTGACCTCTGCACACATG 3' | Genotyping *Pcm1*$^{SNAP}$ mouse. PCR. Product size: 332 bp, only amplifies in WT |
| Sequence-based reagent | *Pcm1* 2R | Sigma | 5' ACAATCGATGGGAGAGCCTC 3' | Genotyping *Pcm1*$^{SNAP}$ mouse. PCR. Product size: 332 bp, only amplifies in WT |
| Sequence-based reagent | *SNAP* F | Sigma | 5' GGCCTGCACCGTATCATCTT 3' | Genotyping *Pcm1*$^{SNAP}$ mouse. PCR. Product size: 132 bp, only amplifies in mutant |
| Sequence-based reagent | *SNAP* R | Sigma | 5' AAAGTAGGCGTTGAGCCAGG 3' | Genotyping *Pcm1*$^{SNAP}$ mouse. PCR. Product size: 132 bp, only amplifies in mutant |
| Chemical compound, drug | SNAP-Cell 647-SiR | New England Biolabs | | |
| Chemical compound, drug | nocodozole | Sigma | SML1665 | 20 µM |
| Antibody | Acetylated Alpha Tubulin | Sigma | 6-11B-1 T6793 | IF (1:1000–1:2000) |
| Antibody | ANKRD26 | GeneTex | GTX128255 | IF(1:100 MeOH) |
| Antibody | ARL13B | Proteintech Group | 17711-1-AP | IF (1:1000, PFA) |
| Antibody | α-tubulin | Sigma | DM1A | WB (1:1000) |
| Antibody | α-tubulin | Abcam | ab4074 | WB (1:1000) |
| Antibody | CENTRIN | Merck | 20 H5 04-1624 | IF (1:300 MeOH w. PE) |
| Antibody | CENTRIOLIN | Santa Cruz | sc-365521 | IF (1:100 MeOH w PE) |
| Antibody | CENTROBIN | Abcam | Ab70448 | IF (1:100 MeOH) |
| Antibody | CEP131 | Proteintech Group | 25735-1-AP | IF (1:75 MeOH w PE) |
| Antibody | CEP162 | Sigma Prestige | HPA030170 | IF(1:100 MeOH) |
| Antibody | CEP164 | Santa Cruz | sc-240226 | IF(1:100 MeOH) |
| Antibody | CEP290 | Santa Cruz | B-7 sc-390462 | IF (1:500 MeOH) |
| Antibody | CEP97 | Proteintech Group | 22050-1-AP | IF(1:100 MeOH) |
| Antibody | CP110 | Proteintech Group | 12780-1-AP | IF/WB (1:1000) |
| Antibody | CP110 | Millipore | MABT1354 | IF (1:100 MeOH w. PE) |
| Antibody | FBF1 | Proteintech Group | 11531-1-AP | IF(1:100 MeOH) |
| Antibody | FOP | Proteintech Group | 11343-1-AP | IF (1:100 PFA or MeOH) |
| Antibody | Gamma Tubulin | Sigma | GTU88 T6557 | IF (1:500, MeOH w PE) |
| Antibody | GAPDH | Proteintech Group | 6008-1-Ig | WB (1:100,000) |

*Appendix 1 Continued on next page*

*Appendix 1 Continued*

| Reagent type (species) or resource | Designation | Source or reference | Identifiers | Additional information |
|---|---|---|---|---|
| Antibody | IFT81 | Proteintech Group | 11744-1-AP | IF (1:100 PFA) |
| Antibody | IFT88 | Proteintech Group | 13967-1-AP | IF (1:100 PFA) |
| Antibody | MIB1 | Sigma | M5948 | IF (1:1000 MeOH w. PE) |
| Antibody | MYOVA | Cell Signaling Technology | 3402S | IF(1:100 MeOH) |
| Antibody | NINEIN | Michel Bornens | L79 | IF(1:200 MeOH) |
| Antibody | PCM1 | Proteintech Group | 19856-1-AP | IF (1:100, MeOH w PE) |
| Antibody | PCM1 C | Novus Biologicals | NBP1-87196 | WB (1:1000) |
| Antibody | PCM1 N | Novus Biologicals | H0005108-B01P | WB (1:1000) |
| Antibody | PCM1 | Santa Cruz | D-19 sc-50164 | (*Figure 7—figure supplement 1*) IF (1:1000 MeOH) |
| Antibody | PCNT | Abcam | ab4448 | IF (1:1000, MeOH) |
| Antibody | Polyglutamylated tubulin | Adipogen HPA030170 | AG-20B-0020-C100/GT335 | IF (1:500) |
| Antibody | RAB34 | Proteintech Group | 27435-1-AP | IF (1:500) |
| Antibody | RPGRIP1L | Proteintech Group | 29778-1-AP | IF (1:100 PFA w 1% SDS) |
| Antibody | TALPID3 | Proteintech Group | 24421-1-AP | IF(1:100 MeOH) |
| Antibody | TTBK2 | Sigma | HPA018113 | IF(I:100) |
| Antibody | ECL -Mouse IgG, HRP-conjugated Host: Sheep | GE Healthcare UK Ltd | | WB (1:7500) |
| Antibody | ECL -Rabbit IgG, HRP-conjugated Host: Sheep | GE Healthcare UK Ltd | | WB (1:7500) |
| Antibody | HRP-conjugated –Rabbit IgG H+L Host: Goat | Bio-Rad | | WB (1:5000) |
| Antibody | HRP-conjugated –Mouse IgG H+L Host: Goat | Bio-Rad | | WB (1:5000) |
| Antibody | Alexa 488-conjugated – Mouse Host: Donkey | Invitrogen Molecular Probes | | IF (1:500) |
| Antibody | Alexa 594-conjugated – Rabbit Host: Donkey | Invitrogen Molecular Probes | | IF (1:500) |
| Antibody | Alexa 488-conjugated – Rabbit Host: Donkey | Invitrogen Molecular Probes | | IF (1:500) |
| Antibody | Alexa 594-conjugated – Mouse Host: Donkey | Invitrogen Molecular Probes | | IF (1:500) |
| Antibody | Alexa 647-conjugated – Rabbit Host: Donkey | Invitrogen Molecular Probes | | IF (1:500) |
| Antibody | Alexa 647-conjugated – Mouse Host: Donkey | Invitrogen Molecular Probes | | IF (1:500) |

*Appendix 1 Continued on next page*

*Appendix 1 Continued*

| Reagent type (species) or resource | Designation | Source or reference | Identifiers | Additional information |
|---|---|---|---|---|
| Antibody | Alexa 647-conjugated – Goat Host: Donkey | Invitrogen Molecular Probes | | IF (1:500) |
| Software algorithm | QuPath | PMID:29203879 | | https://github.com/IGC-Advanced-Imaging-Resource/Hall2022_Paper |
| Software algorithm | Nis-Elements AR V4.6 | Nikon Instruments | | |
| Software algorithm | FIJI | *Schindelin et al., 2012* | | https://github.com/IGC-Advanced-Imaging-Resource/Hall2022_Paper |
| Software algorithm | CellProfiler | *Stirling et al., 2021* | | https://github.com/IGC-Advanced-Imaging-Resource/Hall2022_Paper |
| Software algorithm | Imaris | Oxford Instruments | | |

