## [Editor Report]

This manuscript will be of interest to centrosome and cilia cell biologists and evaluates the in vivo and in vitro role of PCM1, and by extension, centriole satellites in ciliogenesis. The major strength of this study is the detailed characterization of *Pcm1*^−/−^ mutant mice, which reveals a role for PCM1 in the biogenesis of specific types of cilia, such as motile cilia on ependymal cells, by a mechanism involving CP110 and CEP97. The claims are generally well supported by the data.

---

## [Decision Letter]

**Decision letter after peer review:**

Thank you for submitting your article "Centriolar satellites expedite mother centriole remodeling to promote ciliogenesis." for consideration by *eLife*. Your article has been reviewed by 3 peer reviewers, including Lotte B Pedersen as Reviewing Editor and Reviewer #1, and the evaluation has been overseen by Piali Sengupta as the Senior Editor.

Essential revisions:

1) The multiciliated cell phenotype of the Pcm1-/- mice needs to be characterized in more detail to confirm the original results and to be put into the context of the previous study done in vitro (Zhao et al., 2021). Specifically:

(i) The claim that the basal bodies produced in absence of PCM1 show a problem of rotational polarity is not fully supported by the data. To confirm this observation, the authors should look at later time points as P3 is very early and the rotational polarity is progressively established after BB docking and the beginning of cilia beating. Also, many more cells should be analyzed. Since this is a lot of work by EM, one should consider doing it by immunostainings as done in some other papers. Same comment for the absence of ciliary pocket in PCM1 KO. P3 is too early and since some cilia do not show a clear ciliary pocket, one should look in a sufficient number of EM sections.

(ii) The defect in translational polarity is interesting and has never been described before. This phenotype is analyzed at P5 and should also be confirmed at later time point since the delay in multiciliation in the PCM1 KO may affect the number of cells with a terminal differentiated state and therefore bias the result. In fact, migration of BB is the last event occurring during multiciliation.

(iii) The phenotype of cilia beating uncoordination is convincing and confirms what has been also described by Zhao et al., in 2021. It is unclear whether the authors propose a causal link between this beating phenotype in ependymal cells and the proteomic study between WT and PCM1 KO in mTEC. Since the experiments are done in 2 different cell types and the proteomic phenotype resolves in mTEC at ALID21, it is difficult to link the two observations. One would like to know if motility is also affected in mTEC, and to use the proteomic study as an additional explanation for cilia beating alteration, to the one proposed by Zhao et al. (e.g: deregulation of the centriolar/ciliary targeting of satellites client proteins under PCM1 depletion). The structural defects of cilia seen by the authors and by Zhao et al., are also one important piece of explanation.

(iv) in vitro, MCC in PCM1 KO seem to display less cilia. Is this true in vivo in the brain? Since it is not obvious in vivo in the trachea, it would be nice to just address qualitatively whether this is the case in vivo in the brain. Also, are the number of BB affected? Zhao et al., counted the number of BB in PCM1 siRNA treated cells and show no difference. If one would address how PCM1 affect the number of cilia, this is important to know whether less centrioles are produced or whether they fail to dock correctly at the plasma membrane. Since formation of the preciliary vesicle is affected in in RPE1 cells, it is tempting to speculate that a similar defect could arise in MCC and affect motile ciliogenesis. If the « number of cilia » phenotype is not true in vivo, one should also consider a cell culture artefact.

(v) It has been proposed that loss of PCM1 disrupts FGM cores that concentrate centriole proteins near deuterosomes in multiciliated cells undergoing centriole amplification (Zhao et al., 2021). Can the authors stain for components of FGM cores in ependymal cells and mTECs? Could differences in FGM disruption be a possible explanation for the delayed ciliogenesis in ependymal cells but not mTECs? Do PCM1+/+ and PCM1-/- mTECs have similar levels of CP110 on the amplified centrioles?

2) The in vivo relevance of the ciliogenesis defect observed in the RPE1 PCM1-/- cells needs to be clarified and/or discussed more thoroughly. The phenotypes observed in this cell line are likely not occurring in most cell types in vivo, or else the expected organismal phenotypes would probably be even more severe. To address this point, the authors should consider investigating if RPE1 cells in the eye of the Pcm1-/- mouse have a ciliogenesis defect. If so, this could form a nice link that would smooth the transition to RPE1 cell biology. It would also be valuable to stain for (primary) cilia in various tissues from Pcm1-/- mice and characterize if there are changes in cilia length or number. This would be especially interesting in tissues that showed a phenotype from PCM1 loss. For example, some basic characterization of cilia in the kidney would be informative, given the hydronephrosis phenotype.

3) The claim that PCM1 affects ciliogenesis in RPE1 cells by "wicking away" CP110 from the mother centriole is not fully supported by the data provided. The authors need to provide causal evidence to show that a deficiency in the removal of CP110 and CEP97 is responsible for driving the ciliogenesis defect in RPE1 PCM1-/- cells. To this end, the authors could (partially) deplete CP110 from PCM1-/- RPE1 cells to investigate if this rescues the ciliogenesis phenotype of the mutant cells, e.g. as done recently by Goncalves et al. for CEP78-/- cells.

4) The observed requirement for PCM1 in promoting ciliogenesis in RPE1 cells and not MEFs is puzzling, given that the authors still observed increased CP110 levels at the mother centriole in the Pcm1-/- MEFs. In the discussion (lines 464-473), the authors suggest that CP110 removal from the mother centriole may be more important for ciliogenesis in cells using the "extracellular" pathway of ciliogenesis compared to cells forming cilia via the "intracellular" pathway. However, mouse fibroblasts and RPE1 cells were shown to both form cilia via the "intracellular" pathway (e.g. see Ganga et al. 2021) thus this explanation seems insufficient to explain the observed differences between RPE1 cells and MEFs lacking PCM1. Therefore, the authors should consider/discuss alternative mechanisms by which PCM1 might affect ciliogenesis in RPE1 cells, including possibly affecting ciliary targeting of other CP110 regulators such as EDD1/UBR5 (Goncalves et al., 2021), LUBAC and PRPF8 (Shen et al., 2021), and WDR8 knock down (Kurtulmus et al., 2016). Furthermore, if additional clones of PCM1-/- RPE1 cells are available, the authors could quantify cilia frequency/length in these to rule out that the observed phenotype is clone-specific. The authors could also reproduce the MyoVa staining in MEFs to detect whether, in cells forming cilia in the absence of PCM1, the ciliary vesicles are forming properly.

*Reviewer #1 (Recommendations for the authors):*

Figure 1A: please indicate in the figure or legend what the difference is between lanes 2 and 4 (I assume the samples are from the two different Pcm1 mutants?).

Figure 3 supplement 2: the elongated FOP- and centrin-positive structures observed in the Pcm1-/- ependymal cells are puzzling. Since previous work by Spektor et al. (2007) indicated that depletion of CEP97 or CP110 in cells forming primary cilia leads to formation of elongated centrin-positive centrioles, can the authors speculate on how accumulation of CP110 at centrioles in Pcm1-/- ependymal cilia (Figure 6D, I) can lead to an essentially similar phenotype?

Figure 4S1: panels A and B seem redundant as similar data is already shown in Figure 4I, K.

Line 1188: Kumar et al. reference should be updated.

*Reviewer #2 (Recommendations for the authors):*

– Some have observed that clonal selection of RPE1 cells can produce clones with defects in ciliogenesis. How many RPE1 PCM1-/- clones did the authors examine? Can the authors show the phenotypes observed can be rescued by re-expression of a sgRNA resistant PCM1 cDNA?

– All of the experiments in RPE1 PCM1-/- cells report corelative phenotypes. The manuscript would be significantly strengthened if the authors could demonstrate that these changes are causative of the ciliogenesis defects. For example, does CP110 depletion increase ciliation efficiency in PCM1-/- RPE1 cells?

– It has been proposed that loss of PCM1 disrupts FGM cores that concentrate centriole proteins near deuterosomes in multiciliated cells undergoing centriole amplification (Zhao et al., 2021). Can the authors stain for components of FGM cores in ependymal cells and mTECs? Could differences in FGM disruption be a possible explanation for the delayed ciliogenesis in ependymal cells but not mTECs?

– The PCM1-/- mTECs should be characterized a bit more since it is possible that subtle phenotypes may have been missed. The authors could count centriole numbers in mTECs, and quantify the percentage of both MBB stage cells and multiciliated cells. The authors see a delay in the expression of ciliary and motor proteins in PCM1-/- mTECs, so perhaps centrioles are amplifying with normal kinetics but are delayed in docking or ciliating.

*Reviewer #3 (Recommendations for the authors):*

– References to the figures are not in the right order which makes the reading difficult.

– It would be easier to read if the numbering of the figures and supplementary figures was indicated on the pdf file.

– dT are not indicated in the movies.

– Conclusions in the legends should be removed and information such as ages/developmental stages of samples and number of cells analyzed (especially for TEM) should be added.

– Also I haven't find the age at which cilia motility was assessed in ependymal cilia.

– Line 100: it would be more accurate to replace « most » by « half ».

– Line 130, Figure 2K: localisation of the inset should be indicated. Also, what the inset is illustrating is unclear.

– Line 131, Figure Sup2C: for a non-expert, the phenotype described is not visible in the pictures provided.

– Fig3Sup1B: maybe another picture would be better to show that P16 MCC epithelia are comparable between WT and PCM1.

– In PCM1 KO MEFs, *MIB1* satellites does not seem absent to me as indicated line 198 but more scattered. Same for Cep290 in RPE1 cells: satellites seem decreased and scattered but not absent as indicated line 202. Why did the authors choose *Mib1* for one cell type and Cep290 for the other?

– The number of RPE1 ciliated cells does not seem to increase significantly over time in PCM KO. If this is the case, I would remove the sentence line 220.

---

## [Author Response]

Essential revisions:1) The multiciliated cell phenotype of the Pcm1-/- mice needs to be characterized in more detail to confirm the original results and to be put into the context of the previous study done in vitro (Zhao et al., 2021). Specifically:(i) The claim that the basal bodies produced in absence of PCM1 show a problem of rotational polarity is not fully supported by the data. To confirm this observation, the authors should look at later time points as P3 is very early and the rotational polarity is progressively established after BB docking and the beginning of cilia beating. Also, many more cells should be analyzed. Since this is a lot of work by EM, one should consider doing it by immunostainings as done in some other papers. Same comment for the absence of ciliary pocket in PCM1 KO. P3 is too early and since some cilia do not show a clear ciliary pocket, one should look in a sufficient number of EM sections.

We thank the reviewers, and agree the data regarding rotational symmetry defects of basal bodies in *Pcm1* mutant ependymal cells and analysis of ciliary pocket presence are too preliminary to include in this publication. As the comments on rotational symmetry and ciliary pocket presence/absence are not germane to the central points of this work, we removed panels from Figure 3 and Figure 3 —figure supplement 1 and the associated text.

(ii) The defect in translational polarity is interesting and has never been described before. This phenotype is analyzed at P5 and should also be confirmed at later time point since the delay in multiciliation in the PCM1 KO may affect the number of cells with a terminal differentiated state and therefore bias the result. In fact, migration of BB is the last event occurring during multiciliation.

In accordance with the reviewers’ comments, we confirmed that the defect in the translational positioning of basal bodies persists in *Pcm1* mutant cells at P16. These new data affirming that PCM1 promotes the translational polarity of ependymal basal bodies are now included in the updated Figure 3H and Figure 3 —figure supplement 1B and the accompanying text.

(iii) The phenotype of cilia beating uncoordination is convincing and confirms what has been also described by Zhao et al., in 2021. It is unclear whether the authors propose a causal link between this beating phenotype in ependymal cells and the proteomic study between WT and PCM1 KO in mTEC. Since the experiments are done in 2 different cell types and the proteomic phenotype resolves in mTEC at ALID21, it is difficult to link the two observations. One would like to know if motility is also affected in mTEC, and to use the proteomic study as an additional explanation for cilia beating alteration, to the one proposed by Zhao et al. (e.g: deregulation of the centriolar/ciliary targeting of satellites client proteins under PCM1 depletion). The structural defects of cilia seen by the authors and by Zhao et al., are also one important piece of explanation.

We concur with the reviewers that the cilia beating phenotype in *Pcm1* mutant ependymal cells is convincing. To clarify, we are not proposing that the altered ciliary protein composition (assessed using mTEC proteomics) is causal to the ciliary beating phenotypes (observed in ependymal cells). We have included these two datasets to highlight that loss of PCM1 affects both ependymal and tracheal cilia, but differently.

In the revised manuscript, we have, as requested by the reviewers, extended our analysis of ciliary beat to both control and *Pcm1* mutant mTECs and wholemount trachea. In these airway multiciliated cells, we did not detect any PCM1-dependent alteration to ciliary beating. Interestingly, an examination of airway multiciliated cell axonemal structure revealed that they do not display the same structural defects as ependymal axonemes in the absence of PCM1, despite their transiently altered protein composition. Thus, while PCM1 is expressed in both lung and ependymal multiciliated cells, the dependency of these two related cell types on PCM1 for structure and ciliary beating differ. These new data are now included in Figure 3 —figure supplement 3C-F and associated text.

Tracheal and ependymal multiciliated cells exhibit differences in their subcellular organization. For example, the basal bodies of ependymal cells display translational polarization, whereas the basal bodies of tracheal multiciliated cells do not. Perhaps these differences underpin the different requirements for PCM1 in these two multiciliated cell types.

(iv) in vitro, MCC in PCM1 KO seem to display less cilia. Is this true in vivo in the brain? Since it is not obvious in vivo in the trachea, it would be nice to just address qualitatively whether this is the case in vivo in the brain. Also, are the number of BB affected? Zhao et al., counted the number of BB in PCM1 siRNA treated cells and show no difference. If one would address how PCM1 affect the number of cilia, this is important to know whether less centrioles are produced or whether they fail to dock correctly at the plasma membrane. Since formation of the preciliary vesicle is affected in in RPE1 cells, it is tempting to speculate that a similar defect could arise in MCC and affect motile ciliogenesis. If the « number of cilia » phenotype is not true in vivo, one should also consider a cell culture artefact.

We agree that the number of cilia formed per cell is an interesting question, however it is technically difficult to count individual ependymal cilia in vivo. We had already shown in vivo a transient decrease in cells with multiple basal bodies (Figure 3C) and percentage of ependymal cells with multiple cilia (Figure 3D) at P3 and P5. In this revised manuscript, we additionally now counted basal body numbers per cell at P5, and saw no difference in the number of basal bodies per cell in the absence of PCM1 (Figure 3E), although we see more *Pcm1* null ependymal cells with early rosette structures in vivo at the same stage (Figure 3F). We elaborate on the data in cultured ependymal cells, counting basal body number and cilia number throughout differentiation and show a delay in centriole and cilia biogenesis in cultured *Pcm1* ependymal cells (Figure 3 —figure supplement 1C-G), consistent with the delay seen in vivo. We argue this is not cell culture artefact, and the delay in centriole and cilia biogenesis is true both in vivo and in culture, and postulate this underlies the hydrocephaly we observe in *Pcm1^-/-^* mice.

(v) It has been proposed that loss of PCM1 disrupts FGM cores that concentrate centriole proteins near deuterosomes in multiciliated cells undergoing centriole amplification (Zhao et al., 2021). Can the authors stain for components of FGM cores in ependymal cells and mTECs? Could differences in FGM disruption be a possible explanation for the delayed ciliogenesis in ependymal cells but not mTECs? Do PCM1+/+ and PCM1-/- mTECs have similar levels of CP110 on the amplified centrioles?

We agree that how PCM1 loss affects the fibrogranular material (FGM) is an interesting question. To address the questions raised by the reviewers, we stained ependymal cells for FGM components CEP131 and PCNT and mTECs for CEP131. We found that without PCM1, CEP131 shows altered subcellular localisation in both cell types. More specifically, in *Pcm1^-/-^* mTECs, the FGM pool of CEP131 is absent and CEP131 accumulates at the basal bodies, and in *Pcm1^-/-^* ependymal cells, the FGM pool of CEP131 persists, but is altered, displaying a more fibrous organization. In contrast, PCNT subcellular localisation is unaffected in the absence of PCM1. We conclude that the FGM is perturbed in the absence of PCM1 (Figure 3O, Figure 3 —figure supplement 1 H-I). As suggested by the reviewers, this disruption of FGM may contribute to the delayed centriole biogenesis and ciliogenesis observed in the absence of PCM1. As PCM1 is involved in the organization and composition of FGM in both ependymal and tracheal multiciliated cells, it is unclear whether differential effects on FGM underlie differences in the PCM1dependency of ciliogenesis in these two cell types. We now include comment on the PCM1-and cell type-dependent effects on FGM in the results and Discussion sections.

In addition, we analyzed CP110 levels in tracheas. As compared to wild-type siblings, CP110 levels from *Pcm1^-/-^* tracheas were increased (Figure 6 —figure supplement 1). This result corresponds to prior observations of CP110 levels in *Pcm1^-/-^* ependymal cells, RPE1 cell and MEFs. Thus, across diverse cell types, with differing roles for PCM1 in ciliogenesis, PCM1 has a shared requirement in restricting CP110 levels.

2) The in vivo relevance of the ciliogenesis defect observed in the RPE1 PCM1-/- cells needs to be clarified and/or discussed more thoroughly. The phenotypes observed in this cell line are likely not occurring in most cell types in vivo, or else the expected organismal phenotypes would probably be even more severe. To address this point, the authors should consider investigating if RPE1 cells in the eye of the Pcm1-/- mouse have a ciliogenesis defect. If so, this could form a nice link that would smooth the transition to RPE1 cell biology. It would also be valuable to stain for (primary) cilia in various tissues from Pcm1-/- mice and characterize if there are changes in cilia length or number. This would be especially interesting in tissues that showed a phenotype from PCM1 loss. For example, some basic characterization of cilia in the kidney would be informative, given the hydronephrosis phenotype.

While the RPE1 cell line provides a well-characterized model of ciliogenesis, its physiological relevance, as with many cell lines, remains unclear. RPE1 cells are only distantly related to retinal pigmented epithelium in vivo: they are neither pigmented under ciliogenic conditions nor form the cobblestone-like organization characteristic of RPE (Bharti et al., 2022). Because the reviewers asked us to consider investigating the RPE of *Pcm1*^-/-^ mice, we investigated the cilia of RPE both in vivo and as ex vivo cultures. Cultured primary RPE cells were outcompeted by fibroblasts. In flat-mounts from eyes, removing the retina disrupted the RPE primary cilia (as reported in the literature May-Simera et al., 2018). These technical difficulties prevented us from directly addressing these questions in the time given for review. However, we emphasize that, as documented in Figure 2 —figure supplement 1, *Pcm1^-/-^* mice do not show structural or functional defects of the eye, strongly suggesting that ciliogenesis is unaffected in the retinal pigmented epithelium (Sun et al., 2021)*.*

To help address the reviewers’ point about the relevance of ciliogenesis defects detected in *PCM1*^/-^ RPE1 cells to in vivo cells, we expanded our analysis of primary cilia during development. More specifically, we examined cilia in E18.5 embryos, focusing on analysis of the skeleton, kidney and trachea, as these tissues are affected in canonical ciliopathies. In these tissues, we observed no PCM1-dependent differences in the number, length or morphology of cilia (Figure 1 —figure supplement 3). Our analysis does not preclude functional defects in the absence of PCM1 and emphasizes the tissue-specific effects of PCM1 loss on ciliogenesis. A commonality between *PCM1*^-/-^ RPE1 cells and diverse other cell types, including MEFs, ependymal cells, and tracheal multiciliated cells, is that PCM1 promotes the removal of CP110 from distal mother centrioles, an early step in ciliogenesis. We propose that while RPE1 cells are imperfect models of the in vivo retinal pigmented epithelial cell layer, their utility lies in their assistance in discovering molecular alterations that depend on PCM1, such as efficient removal of CP110, an early step in ciliogenesis.

3) The claim that PCM1 affects ciliogenesis in RPE1 cells by "wicking away" CP110 from the mother centriole is not fully supported by the data provided. The authors need to provide causal evidence to show that a deficiency in the removal of CP110 and CEP97 is responsible for driving the ciliogenesis defect in RPE1 PCM1-/- cells. To this end, the authors could (partially) deplete CP110 from PCM1-/- RPE1 cells to investigate if this rescues the ciliogenesis phenotype of the mutant cells, e.g. as done recently by Goncalves et al. for CEP78-/- cells.

Our data suggest that centriolar satellites help mediate two closely related steps in ciliogenesis: ciliary vesicle formation and removal of CP110 and CEP97. Based on the reviewer’s suggestions, we assessed whether depletion of CP110 would rescue the ciliogenesis defect of *PCM1*^-/-^ RPE1 cells, similar to the approach used expertly by Goncalves et al. We used two different shRNAs to partially deplete CP110 in wild-type and *PCM1*^-/-^ RPE1 cells. In contrast to the aforementioned work, in our hands, CP110 loss attenuated ciliogenesis in wild-type RPE1 cells (Author response image 1). Perhaps consequently, loss of CP110 did not rescue the ciliogenesis defect in *PCM1*^-/-^ RPE1 cells. One reason for the difference of results may be the degree of depletion. It is possible that a more moderate degree of knockdown would show a different result.

**Author response image 1. sa2fig1:** (**A**) Wild-type and *PCM1*-/- RPE1 cells were transduced with scrambled (Scr) control or CP110 shRNAs (#1 and #2). Cells were serum starved for 24 h and immunostained with ARL13B (magenta) and FOP (yellow). (**B**) Immunoblotting showed that the two CP110 shRNAs reduced CP110 protein levels. (**C**) Quantification of ciliation frequency 24 h post serum deprivation after depletion of CP110. Bar graphs show means ± SEM. Two-way Anova: *** p < 0.001 and ns is not significant. n>100 cells from 2 replicates.

4) The observed requirement for PCM1 in promoting ciliogenesis in RPE1 cells and not MEFs is puzzling, given that the authors still observed increased CP110 levels at the mother centriole in the Pcm1-/- MEFs. In the discussion (lines 464-473), the authors suggest that CP110 removal from the mother centriole may be more important for ciliogenesis in cells using the "extracellular" pathway of ciliogenesis compared to cells forming cilia via the "intracellular" pathway. However, mouse fibroblasts and RPE1 cells were shown to both form cilia via the "intracellular" pathway (e.g. see Ganga et al. 2021) thus this explanation seems insufficient to explain the observed differences between RPE1 cells and MEFs lacking PCM1. Therefore, the authors should consider/discuss alternative mechanisms by which PCM1 might affect ciliogenesis in RPE1 cells, including possibly affecting ciliary targeting of other CP110 regulators such as EDD1/UBR5 (Goncalves et al., 2021), LUBAC and PRPF8 (Shen et al., 2021), and WDR8 knock down (Kurtulmus et al., 2016).

We concur that potential alternate mechanisms may explain cell type-dependent effects on ciliogenesis better than dependence on the intracellular and extracellular ciliogenic pathway. Therefore, we have expanded our Discussion with a section on how centriolar satellites may help deliver proteins such as EDD1/UBR5, LUBAC and PRPF8 to alter CP110 and CEP97 levels at the mother centriole.

Furthermore, if additional clones of PCM1-/- RPE1 cells are available, the authors could quantify cilia frequency/length in these to rule out that the observed phenotype is clone-specific.

To exclude clone-specific results in *PCM1*^-/-^ RPE1 cells, we quantified ciliogenesis frequency from two additional clonal *PCM1*^-/-^ RPE1 cell lines (Figure 4J). Across all *PCM1*^-/-^ RPE1 cell lines, we observed decreased ciliation frequency. We also performed rescue experiments by overexpressing eYFP-PCM1 in two independent clonal *PCM1*^-/-^ RPE1 cell lines and observed rescue of the ciliogenesis phenotype (Figure 4L).

The authors could also reproduce the MyoVa staining in MEFs to detect whether, in cells forming cilia in the absence of PCM1, the ciliary vesicles are forming properly.

As suggested by the reviewer, we attempted to immunostain MEFs with Myosin-Va (that marks preciliary and ciliary vesicles) as well as a marker of ciliary vesicles, RAB34. Both antibodies recognize the human epitopes in RPE1 cells (RAB34 staining in RPE1 cells is now included in Figure 5C, D). Neither antibody recognizes the mouse epitopes. Therefore, we were unable to directly assess ciliary vesicle formation in MEFs.

References

Bharti K, den Hollander AI, Lakkaraju A, Sinha D, Williams DS, Finnemann SC, Bowes-Rickman C, Malek G, D’Amore PA. 2022. Cell culture models to study retinal pigment epithelium-related pathogenesis in age-related macular degeneration. Exp Eye Res 222:109170. doi:10.1016/j.exer.2022.109170

Gonçalves AB, Hasselbalch SK, Joensen BB, Patzke S, Martens P, Ohlsen SK, Quinodoz M, Nikopoulos K, Suleiman R, Damsø Jeppesen MP, Weiss C, Christensen ST, Rivolta C, Andersen JS, Farinelli P, Pedersen LB. 2021. CEP78 functions downstream of CEP350 to control biogenesis of primary cilia by negatively regulating CP110 levels. eLife 10. doi:10.7554/eLife.63731

Kurtulmus B, Wang W, Ruppert T, Neuner A, Cerikan B, Viol L, Dueñas-Sánchez R, Gruss OJ, Pereira G. 2016. WDR8 is a centriolar satellite and centriole-associated protein that promotes ciliary vesicle docking during ciliogenesis. J Cell Sci 129:621– 636. doi:10.1242/jcs.179713

May-Simera HL, Wan Q, Jha BS, Hartford J, Khristov V, Dejene R, Chang J, Patnaik S, Lu Q, Banerjee P, Silver J, Insinna-Kettenhofen C, Patel D, Lotfi M, Malicdan M, Hotaling N, Maminishkis A, Sridharan R, Brooks B, Miyagishima K, Gunay-Aygun M,

Pal R, Westlake C, Miller S, Sharma R, Bharti K. 2018. Primary Cilium-Mediated Retinal Pigment Epithelium Maturation Is Disrupted in Ciliopathy Patient Cells. Cell Rep 22:189–205. doi:10.1016/j.celrep.2017.12.038

Sun C, Zhou J, Meng X. 2021. Primary cilia in retinal pigment epithelium development and diseases. J Cell Mol Med 25:9084–9088. doi:10.1111/jcmm.16882